# Molecular basis for PICS-mediated piRNA biogenesis and cell division

Xiaoyang Wang[1,3], Chenming Zeng[1,3], Shanhui Liao [1,3], Zhongliang Zhu[1], Jiahai Zhang[1], Xiaoming Tu[1], Xuebiao Yao [1], Xuezhu Feng[1✉], Shouhong Guang [1,2✉] & Chao Xu [1✉]

By incorporating two mutually exclusive factors, PID-1 and TOST-1, *C. elegans* PICS complex plays important roles in piRNA biogenesis, chromosome segregation and cell division. We firstly map the interaction network between PICS subunits, then uncover the mechanisms underlying the interactions between PICS subunits by solving several complex structures, including those of TOFU-6/PICS-1, ERH-2/PICS-1, and ERH-2/TOST-1. Our biochemical experiment also demonstrates that PICS exists as an octamer consisting of two copies of each subunit. Combining structural analyses with mutagenesis experiments, we identify interfacial residues of PICS subunits that are critical for maintaining intact PICS complex in vitro. Furthermore, using genetics, cell biology and imaging experiments, we find that those mutants impairing the in vitro interaction network within PICS, also lead to dysfunction of PICS in vivo, including mislocalization of PICS, and reduced levels of piRNAs or aberrant chromosome segregation and cell division. Therefore, our work provides structural insights into understanding the PICS-mediated piRNA biogenesis and cell division.

[1] Ministry of Education Key Laboratory for Membraneless Organelles & Cellular Dynamics, Hefei National Laboratory for Physical Sciences at the Microscale, School of Life Sciences, Division of Life Sciences and Medicine, University of Science and Technology of China, Hefei, PR China. [2] Department of Obstetrics and Gynecology, The First Affiliated Hospital of USTC, Division of Life Sciences and Medicine, University of Science and Technology of China, Hefei, PR China. [3] These authors contributed equally: Xiaoyang Wang, Chenming Zeng, Shanhui Liao. ✉email: fengxz@ustc.edu.cn; sguang@ustc.edu.cn; xuchaor@ustc.edu.cn

Piwi-interacting RNAs (piRNA) associate with PIWI family proteins, and play essential roles in transposon silencing and gene regulation[1–5]. PIWI proteins were firstly discovered as essential factors for stem cell self-renewal in *Drosophila*[6]. Lack of PIWI proteins also leads to male sterile in mice and human[7–10]. In *C. elegans*, Piwi-like protein PRG-1 loaded piRNAs recognize respective target sequences in a manner following the mismatch-tolerant rule between piRNA sequences and their targets[11–18]. Upon targeting, the piRNA/PRG-1 complex recruits RNA-dependent RNA polymerase to drive the biogenesis of secondary small interfering RNA to conduct the silencing of piRNA targets[4,19–24].

Diverged piRNA biogenesis pathways exist in animals[3]. In the germline of *C. elegans*, primary piRNA precursors are transcribed by RNA polymerase II from thousands of piRNA loci containing a core DNA consensus motif, which is termed as "Ruby motif"[25,26]. The upstream sequence transcription complex (USTC) consisting of PRDE-1, SNPC-4, and TOFU-4/5, initiates the transcription of piRNA genes, while the integrator complex and the RNA polymerase II subunit RBP-9 act in the termination of piRNA transcription[27–32]. SNPC-1.3 associates with SNPC-4 and promotes the piRNA transcription in males[33]. After transcription, the precursors are transported into cytoplasm or perinuclear granules and decapped to remove two nucleotides at 5′ ends, allowing them to be loaded by PRG-1[3,19–21]. For the PRG-1-bound piRNA, its extra nucleic acids at 3′ ends are trimmed by the exonuclease, PARN-1[34]. At the final step of piRNA maturation, the 2′-O-methylation at 3′ termini is carried out by HENN-1[35–37]. The mature piRNAs have a uniform length of 21 nt, starting with a 5′ uracil and ending with a 2′-O-methylated 3′ residue, thereby termed as 21U-RNAs[19,20,25].

By using functional proteomics, we and another group independently identified a piRNA biogenesis and chromosome segregation (PICS) complex (also named as PETISCO) that plays important roles in 21U-RNA biogenesis, chromosome segregation and cell division[38,39]. The PICS complex is composed of TOFU-6, PICS-1 (also names as PID-3), ERH-2 and two mutually exclusive factors, PID-1 and TOST-1. The PID-1 containing complex is functional in piRNA maturation and processes piRNA precursors at 5′ termini via its association with IFE-3, while the TOST-1 containing form is essential for chromosome segregation and cell division in embryos. Therefore our work provides a good example by showing that PICS complex adapts to two distinct cellular functions, piRNA biogenesis and cell division, by incorporating two mutually exclusive factors[38–41]. Although IFE-3, which encodes the *C. elegans* ortholog of human eIF4E, binds to the C-terminal region of TOFU-6, named as IFE-3 binding motif (IBM) hereafter[38,39], its depletion does not affect the localization of the other PICS subunits[39]. Previous work indicate that the PICS-1 (also known as PID-3) binds to the RNA-Recognition Motif (RRM) of TOFU-6 and ERH-2, and ERH-2 further binds to TOST-1 or PID-1[38,39]. In addition, PICS was purified from cell extracts as a complex of ~400 kD, suggesting the possibility of multiple copies of each subunit[38]. However, the molecular mechanisms underlying the interaction network within PICS complex are elusive.

To provide structural insights into the PICS complex, we first examined the interactions between PICS subunits using the Isothermal Titration Calorimetry (ITC) binding assay. ITC binding data indicate that the RNA-Recognition Motif (RRM) domain and a fragment upstream of the RRM domain of PICS-1(PID-3) interact with the TOFU-6 RRM domain and ERH-2, respectively, and TOST-1 and PID-1 bind to ERH-2 in a mutually exclusive manner. Then we solved the complex structure of PICS-1$^{RRM}$ with TOFU-6$^{RRM}$, in which the PICS-1$^{RRM}$ homodimer binds two TOFU-6$^{RRM}$ molecules to form a heterotetramer. Next, we

solved the structures of ERH-2 homodimer bound with either a PICS-1(PID-3) fragment or a TOST-1 fragment, and found that PICS-1(PID-3) and TOST-1 binds to different surfaces of ERH-2, whereas PID-1 and TOST-1 bind to ERH-2 in a competitive manner. Our structures support that PICS complex is an octamer formed by two copies of each four subunits. In PICS, one ERH-2 dimer binds to one TOFU-6$^{RRM}$/PICS-1$^{RRM}$ heterotetramer to form a hexamer, and two copies of TOST-1 or PID-1 further bind to the hexamer to form the octamer. We validated the binding interface between PICS subunits by using mutagenesis and ITC binding experiments. Furthermore, we found that the disruption of PICS complex in vivo leads to reduced amount of mature piRNAs or the delay in chromosome segregation during cell division. Overall our structure research, complemented by in vitro biochemistry and in vivo cell biology experiments, not only unveils the molecular mechanism underlying PICS formation, but also sheds light on the role of PICS complex in piRNA maturation and chromosome segregation.

## Results

**Interaction network within PICS mapped by ITC binding assay.** Previously, we and another group identified the PICS complex and mapped the protein-protein interactions (PPIs) by pull-down and western blot experiments (Fig. 1a)[38,39]. Here we cloned, expressed, and purified the RRM domains of TOFU-6 (TOFU-6$^{RRM}$, residues 2-92) and PICS-1(PID-3) (PICS-1$^{RRM}$, residues 201-282), respectively, and examined their binding affinity by ITC. The ITC binding data indicate that the TOFU-6$^{RRM}$ binds to PICS-1$^{RRM}$ with a $K_d$ of 6.2 nM (Fig. 1b, Supplementary Table 1). By ITC binding assay, we found that ERH-2 binds to a region upstream of PICS-1$^{RRM}$ (residues 180-200), named as ERH-2 binding motif (EBM) thereafter, with a $K_d$ of 8.5 μM. In addition, ERH-2 also binds to two fragments derived from PID-1 (PID-1$^{50-70}$) and TOST-1 (TOST-1$^{34-54}$) with $K_d$s of 18 μM and 3.2 μM, respectively (Supplementary Table 1). To increase the solubility of above ERH-2 binding fragments, we fused them to the C-terminus of SUMO protein and the $K_d$s are for the binding between ERH-2 and SUMO fusion proteins. As a negative control, SUMO protein itself displays no binding affinity towards ERH-2 (Supplementary Table 1).

To understand how ERH-2 bind to different ligands in vitro simultaneously, we fused ERH-2 with PICS-1$^{180–200}$ and TOST-1$^{34–54}$ via a 21-aa linker ((Gly-Ser-Ser)$_7$) and an 18-aa linker ((Gly-Ser-Ser)$_6$), respectively. We then examined the binding of PID-1$^{50-70}$ to the two fusion proteins to study whether the ERH-2/PID-1 interaction is affected in the presence of PICS-1(PID-3) or TOST-1. ITC binding data show that PID-1 binds to ERH-2-(GSS)$_7$-PICS-1 but not ERH-2-(GSS)$_6$-TOST-1, with a comparable binding affinity with ERH-2 only (12 μM vs. 18 μM) (Supplementary Table 1), suggesting that TOST-1 but not PICS-1(PID-3) interferes with the *in vitro* interaction between ERH-2 and PID-1. Taken together, the ITC binding experiments suggest that both PICS-1(PID-3) and ERH-2 serve as scaffold molecules to tether different factors within PICS complex.

**Crystal structure of PICS-1$^{RRM}$-TOFU-6$^{RRM}$ tetramer.** To understand how PICS-1$^{RRM}$ interacts with TOFU-6$^{RRM}$, we crystalized the complex and solved the SeMet and native structures at resolutions of 1.95 Å and 2.68 Å, respectively (Table 1). Given that the two structures are almost identical, the native one was used for structural analysis. In the complex, the two PICS-1$^{RRM}$ molecules (PICS-1$^A$ and PICS-1$^B$) a symmetric homodimer (P2), with each of them adopting canonical RRM fold (β1-α1-β2-β3-α2-β4) (Fig. 1c, Supplementary Fig. 1a, b)[42]. Two TOFU-6$^{RRM}$ molecules (TOFU-6$^C$ and TOFU-6$^D$) interacts with P2 in a

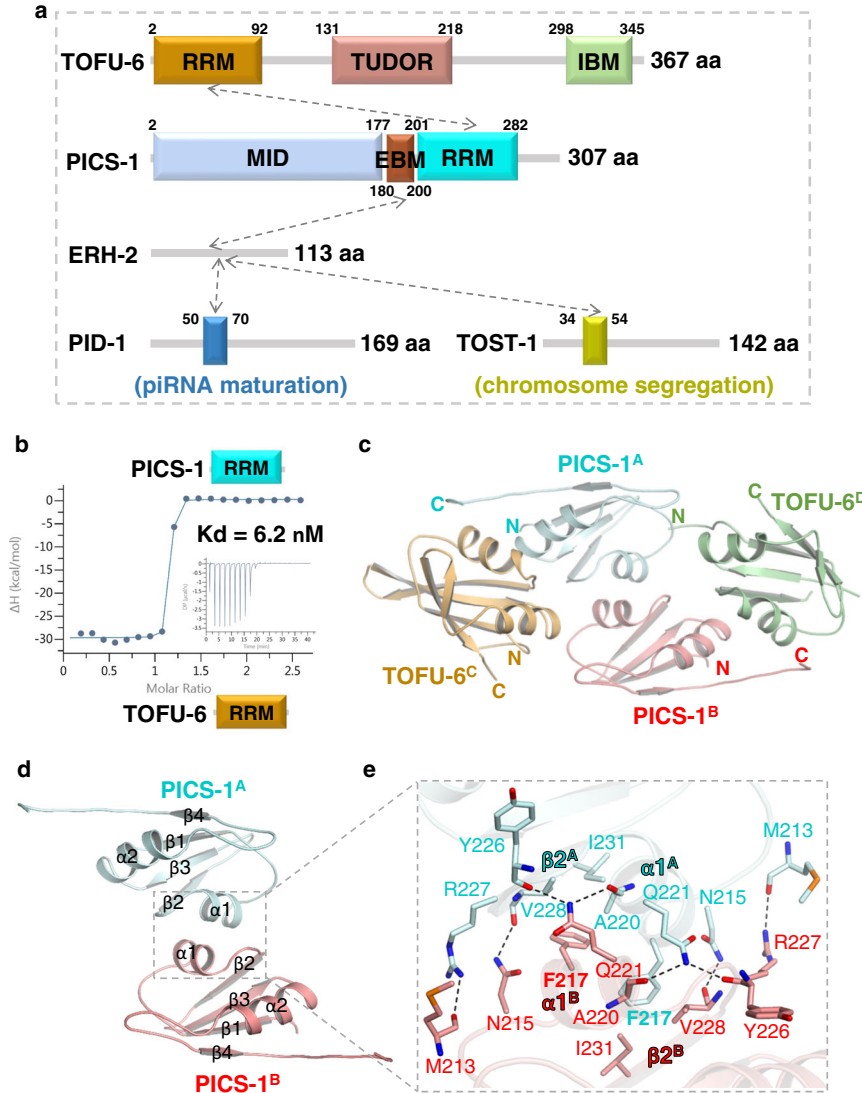

**Fig. 1 Interaction network within PICS complex. a** Domain architecture of PICS subunits, TOFU-6, PICS-1(PID-3), ERH-2, and TOST-1 or PID-1. Domain boundaries are numbered and interactions between PICS subunits are indicated by grey dash arrows. IBM: IFE-3 binding motif. Functions of TOST-1- and PID-1- containing complexes are different. **b** ITC binding curves for PICS-1$^{RRM}$ with TOFU-6$^{RRM}$. **c** Overall structure of TOFU-6$^{RRM}$/ PICS-1$^{RRM}$ heterotetramer. Two PICS-1$^{RRM}$ molecules are shown in red and cyan cartoon, respectively, while the two TOFU-6$^{RRM}$ molecules are shown in orange and green cartoon, respectively. **d** Interface of PICS-1$^{RRM}$ homodimer. Two PICS-1$^{RRM}$ molecules are shown in the same way as in Fig. 1c. **e** Detailed interactions between two PICS-1$^{RRM}$ protomers in the homodimer. Residues involved in intermolecular interactions are shown in sticks with hydrogen bonds shown in black dashes.

symmetric manner, with each TOFU-6$^{RRM}$ contacting with both protomers of P2 to form the TOFU-6/PICS-1 tetramer (T2-P2) (Fig. 1c).

The two PICS-1$^{RRM}$ interact with each other through α1, β2, and the linker connecting α1 and β2, with α1$^A$ nearly perpendicular to α1$^B$ (Fig. 1d, e). For one half of the interface, Phe217$^A$ is buried in a hydrophobic pocket composed of Phe217$^B$, Ala220$^B$, V228$^B$, and I231$^B$ (Fig. 1e). In addition, several hydrogen bonds are formed, including one between main chain carboxyl group of Met213$^A$ and the side chain of Arg227$^B$, and one between the side chain of Asn215$^A$ and the main chain carboxyl group of Val228$^B$, and two between the side chain amide group of Gln221$^A$ and the main chain carboxyl groups of Ala220$^B$ and Tyr226$^B$ (Fig. 1e). Altogether, both hydrophobic and electrostatic interactions contribute to the dimerization and the other half of the dimerization interface could be obtained via the symmetry. TOFU-6$^{RRM}$ is longer than PICS-1$^{RRM}$, and it

contains five anti-parallel β-strands with β4 as the additional one immediately after α2 (Supplementary Fig. 2a, b). TOFU-6$^C$ primarily interacts with PICS-1$^A$ through α1-β2 of TOFU-6$^C$, and α2 and a C-terminal fragment of PICS-1$^A$ (Fig. 2a–c). α1 of TOFU-6$^C$ interacts with α2 of PICS-1$^A$ via both hydrophobic and electrostatic interactions (Fig. 2b). Specifically, Trp27, Phe30, His31, and Cys42 of TOFU-6$^C$, make hydrophobic interactions with Lys246, Phe247, and Tyr250 of PICS-1$^A$. Asp26 and Gln34 of TOFU-6$^C$ form salt bridges with Lys246 and Gln254 of PICS-1$^A$, respectively (Fig. 2b). In addition, α1 of TOFU-6$^C$ also interacts with the PICS-1(PID-3) C-terminus via electrostatic interactions (Fig. 2b). The imidazole ring of TOFU-6$^C$ His31 hydrogen bonds to the side chain of the PICS-1$^A$ Ser275; the TOFU-6$^C$ Asn25 forms two hydrogen bonds via its side chain with the main chain groups of Ser276 and Ala278 of PICS-1$^A$; the side chain of Asn28 also forms two hydrogen bonds with the main chain groups of Ala278 and Val279 of PICS-1$^A$ (Fig. 2b).

**Table 1 Data collection and refinement statistics.**

| | TOFU-6$^{RRM}$ (L73M)-PICS-1$^{RRM}$ (I233M) (SeMet) | TOFU-6$^{RRM}$-PICS-1$^{RRM}$ (Native) | ERH-2$^{1-103}$-(GSS)$_7$-PICS-1$^{180-200}$ | ERH-2$^{1-103}$-(GSS)$_6$-TOST-1$^{34-54}$ |
|---|---|---|---|---|
| Data collection | | | | |
| Space group | P 2$_1$ 2$_1$ 2$_1$ | P 6$_2$ 2 2 | P 2$_1$ 2$_1$ 2$_1$ | P 2$_1$ 2$_1$ 2 |
| Cell dimensions | | | | |
| *a, b, c* (Å) | 54.69, 86.91, 92.79 | 116.96, 116.96, 140.17 | 44.83, 52.53, 125.12 | 103.12, 51.88, 57.04 |
| α, β, γ (°) | 90, 90, 90 | 90, 90, 120 | 90, 90, 90 | 90, 90, 90 |
| Resolution (Å) | 50–1.95(2.02–1.95)* | 50–2.68(2.78–2.68) | 50.00–2.40(2.44–2.40) | 50–2.20(2.28–2.20) |
| $R_{merge}$ | 0.143(1.075) | 0.039(1.161) | 0.099 (0.654) | 0.074(0.507) |
| *I / σI* | 19.2(2.3) | 59(3.0) | 24.0(2.0) | 34.0(3.5) |
| Completeness (%) | 100(99.8) | 100(100) | 100(99.8) | 100(100) |
| Redundancy | 12.5(9.6) | 37.7(34.8) | 11.6(7.1) | 11.4(8.7) |
| Refinement | | | | |
| Resolution (Å) | 27.65–1.95 | 23.36–2.68 | 26.26–2.40 | 25.94–2.20 |
| No. of reflections | 32442 | 16286 | 11991 | 15179 |
| $R_{work}$ / $R_{free}$ | 0.184/0.225 | 0.195/0.236 | 0.201/0.238 | 0.178/0.224 |
| No. atoms | 2886 | 2616 | 1679 | 1702 |
| Protein | 2569 | 2568 | 1671 | 1628 |
| Ligand/ion | N/A | 10 | N/A | N/A |
| Water | 317 | 38 | 8 | 74 |
| *B*-factors | | | | |
| Protein | 24.8 | 47.0 | 71.2 | 36.1 |
| Ligand/ion | N/A | 77.6 | N/A | N/A |
| Water | 35.4 | 33.9 | 55.8 | 40.2 |
| R.m.s. deviations | | | | |
| Bond lengths (Å) | 0.007 | 0.009 | 0.005 | 0.009 |
| Bond angles (°) | 0.85 | 1.0 | 0.71 | 1.0 |

*Values in parentheses are for highest-resolution shell.

Two loops of TOFU-6$^C$, one between α1 and β2 and the other between β3 and α2, also interacts with PICS-1$^A$ (Fig. 2c). Lys38 and Met61 of Tofu-6$^C$ make hydrophobic contacts with Pro225 and Tyr226 of PICS1$^A$; TOFU-6$^C$ Val39 forms two main chain hydrogen bonds with the side chain of PICS-1$^A$ Gln251, and TOFU-6$^C$ Ser40 also forms two hydrogen bonds via its main chain carboxyl group with the side chain amide groups of Asn245 and Gln248 (Fig. 2c).

TOFU-6$^C$ also contacts with the other protomer of PICS-1(PID-3) (PICS-1$^B$) (Fig. 2d). The N-terminus and α2 of TOFU-6$^C$ interact with two loops of PICS-1$^B$, one between β1 and α1 and the other between α2 and β4. Tyr8 and Leu10 of TOFU-6$^C$ make hydrophobic contacts with Pro211, Met213, and His265 of PICS-1$^B$. Asp12 and Asp67 of TOFU-6$^C$ form salt bridges with Arg263 of PICS-1$^B$. Due to symmetry, TOFU-6$^D$ also interacts with both protomers of P2, primarily via PICS-1$^B$ (Fig. 1c).

To validate the P2 interface that constitutes the basis of the heterotetramer, we chose to mutate Phe217 of PICS-1$^{RRM}$ to Glu and found by gel filtration chromatography that F217E disrupts the PICS-1$^{RRM}$ homodimer by introducing repulsive charges and potential steric clashes (Supplementary Fig. 3a), confirming the key role of Phe217 in the P2 dimer interface. Also, the PICS-1$^{RRM}$ F217E reduced the TOFU-6$^{RRM}$ binding affinity by ~90-fold ($K_d$s: 6.2 nM vs. 570 nM) (Supplementary Fig. 3b, Supplementary Table 1), suggesting that the second interface between P2 and TOFU-6$^{RRM}$ is important for maintaining high binding affinity within the tetramer. Next we chose to mutant residues at the PICS-1$^{RRM}$/TOFU-6$^{RRM}$ interface and examined their effect by ITC binding assays. ITC data indicate all mutants, including Y250A/Q251A and K246A/F247A of PICS-1$^{RRM}$, and D26A/W27A of TOFU-6 $^{RRM}$, reduced the binding affinity by ~2700-4500 folds ($K_d$s: 6.2 nM vs. 17-30 μM) (Fig. 2e-g, Supplementary Table 1), further validating the PICS-1$^{RRM}$/TOFU-6$^{RRM}$ interface. Collectively, our mutagenesis experiments and ITC binding data confirm the roles of residues at P2 dimer and T2-P2 tetramer interfaces, demonstrating that an intact P2 is required for high-affinity interaction with TOFU-6$^{RRM}$.

Homodimerization of PICS-1$^{RRM}$ is similar to those observed for the MEC-8 (PDB ID: 5TKZ) and HuR (PDB ID: 6GC5) RRM homodimers, both of which homodimerize via the α1 helix and have the RNA binding capacity (Supplementary Fig. 4). The RNA-bound complex structures demonstrate that MEC-8$^{RRM}$ and HuR$^{RRM}$ bind their nucleotide ligands in a way common for other known RRM domains (Supplementary Fig. 4c-f). Of note, the corresponding surfaces of PICS-1$^{RRM}$ and TOFU-6$^{RRM}$ are also positively charged (Supplementary Fig. 4a-b), suggesting both of them likely employ similar RNA binding mode.

**Structure of ERH-2 bound to PICS-1$^{EBM}$.** Our ITC data showed that ERH-2 binds to PICS-1$^{EBM}$, named as EBM hereafter (Fig. 3a). To uncover the molecular mechanism underlying ERH-2/EBM interaction, we tried the co-crystallization of ERH-2 with EBM, but failed to get any diffractable crystals. Then we fused a fragment of ERH-2 (1-103) with the EBM of PICS-1(PID-3) (180-200) via a (GSS)$_7$ linker, crystallized the fusion protein, and solved its structure at a resolution of 2.40 Å (Table 1). The overall structure is a tetramer formed by ERH-2 and EBM in a 2:2 ratio. In the structure, the ERH-2 homodimer adopts a butterfly-like architecture with the antiparallel β sheet from each protomer packed against each other (Fig. 3b). The structure of *C. elegans* ERH-2 is similar to its human and yeast orthologs with the root-mean-square deviations (RMSDs) in a range of 1.1-1.3 Å (Supplementary Fig. 5).

Two EBM molecules, named as EBM$^C$ and EBM$^D$, respectively, are visible in the structure with each of them adopting a helix structure (Fig. 3b). The EBM$^C$ is accommodated into a v-shape concave via contacting with both ERH-2 molecules, ERH-2$^A$ and ERH-2$^B$, respectively (Fig. 3b). EBM$^C$ mainly interacts with the β1, β2, α1, and α2 of ERH-2$^A$, and extensive hydrophobic interactions are found throughout their binding interface. EBM$^C$

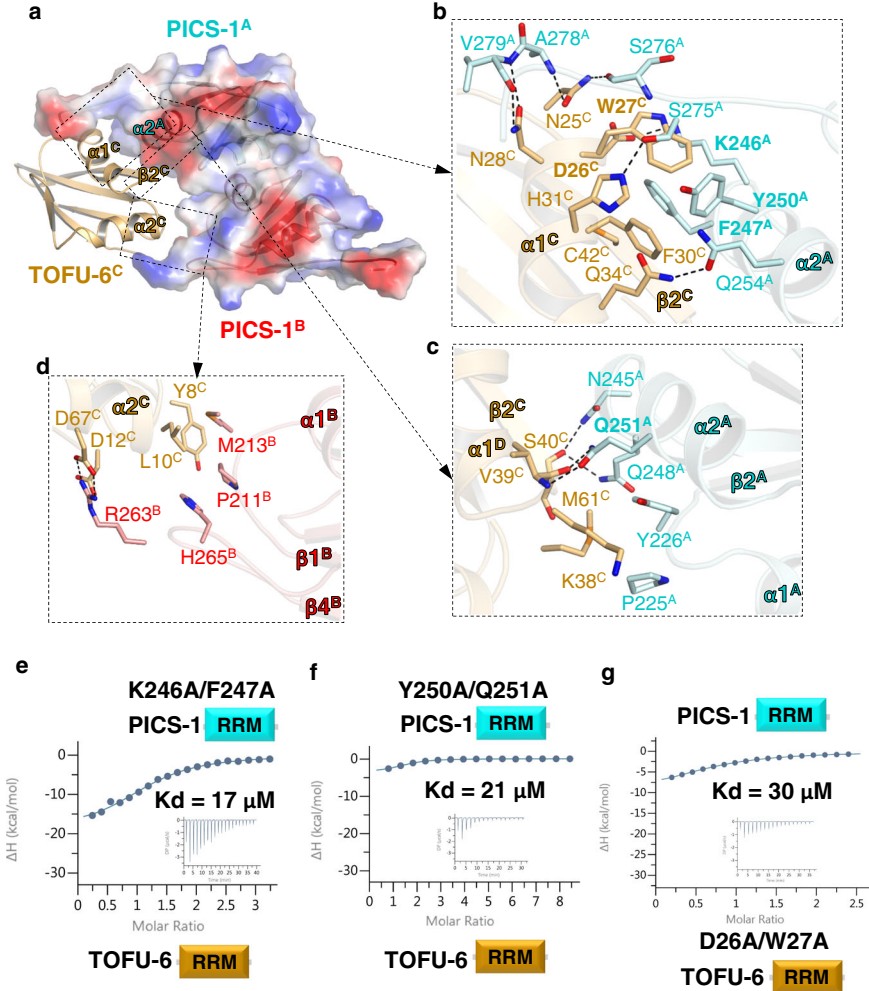

**Fig. 2 The PICS-1$^{RRM}$ dimer bind to Two TOFU-6$^{RRM}$ molecules. a** The electrostatic surface of the PICS-1$^{RRM}$ dimer bound with TOFU-6$^{RRM}$. Due to symmetry, only one TOFU-6$^{RRM}$ molecule, TOFU-6$^{C}$ is shown in orange cartoon. **b**, **c** Detailed interactions between TOFU-6$^{C}$ and one PICS-1(PID-3) protomer, PICS-1$^{A}$. **d** Detailed interactions between TOFU-6$^{C}$ and the other PICS-1(PID-3) protomer, PICS-1$^{B}$. In Fig. 2b–d, residues involved in TOFU-6/ PICS-1 interaction are shown in sticks. Hydrogen bonds are shown in black dashes. **e**, **f** ITC binding curves for mutants of PICS-1$^{RRM}$ with TOFU-6$^{RRM}$. **g** ITC binding curves for PICS-1$^{RRM}$ with TOFU-6$^{RRM}$ mutant.

also interacts with the η1 of ERH-2$^{B}$ via its C-terminus (Fig. 3c). On one side, I182, V186, Phe187, and Leu190 of EBM$^{C}$ make hydrophobic interactions with Ile11, Thr13, Met38, Phe42, Phe61, and Leu65 of ERH-2$^{A}$; on the other side, Val189 of EBM$^{C}$ is accommodated into a hydrophobic pocket composed of Ala34, Lys37, and Met38 of ERH-2$^{A}$ (Fig. 3c). In addition to hydrophobic interactions, the main chain of EBM$^{C}$ Asp181 and the side chain of EBM$^{C}$ Ser185 are hydrogen bonded to the side chain of Lys45$^{A}$; the side chain of EBM$^{C}$ Ser185 forms another hydrogen bond with ERH-2$^{A}$ Asp41; the main chain of EBM$^{C}$ Leu190 is hydrogen bonded to the side chain of ERH-2$^{B}$ Arg20 and the side chain of EBM$^{C}$ His191 form two hydrogen bonds with the side chains of ERH-2$^{A}$ Thr21 and ERH-2$^{B}$ Arg20, respectively. EBM$^{C}$ His191 also makes hydrophobic interaction with Leu17$^{B}$ (Fig. 3c). EBM$^{D}$ contacts with the ERH-2 dimer in a symmetrical manner (Fig. 3b). Taken together, each EBM contacts with both protomers of ERH-2 dimer through both hydrophobic and hydrogen bonding interactions.

To examine the roles of the ERH-2/PICS-1 interface residues, we mutated the EBM residues and examined the effect in ERH-2 binding by ITC. ITC binding data indicate that neither of EBM double mutants, I182D/V186D and V189D/L190D, displayed ERH-2 binding affinity, which highlights the roles of

hydrophobic residues in binding to ERH-2 (Fig. 3d, e, Supplementary Table 1).

**Structure of TOST-1-bound ERH-2.** Consistent with previous work[38,39], our ITC binding data also show that ERH-2 also binds to the fragments of TOST-1 and PID-1 in a mutually exclusive manner (Fig. 4a, Supplementary Table 1), and either of them effectively interacts with ERH-2 in the presence of EBM (Supplementary Table 1). To understand the molecular mechanism how TOST-1 binds to ERH-2, we made a similar fusion protein by linking ERH-2(1–103) with a fragment of TOST-1(34–54) via a (GSS)$_6$ linker, crystallized the fusion protein, and determined its structure at a 2.20 Å resolution (Table 1). In the TOST-1-bound structure, the ERH-2 homodimer is similar to that in the PICS-1-bound one. However, only one TOST-1 molecule (aa 35–54) is visible in the structure since the crystal packing prevents the access of the second TOST-1 to the symmetrical site (Supplementary Fig. 6).

The TOST-1 fragment (TOST-1$^{34–54}$) adopts a helix structure with two extended ends, and the helix is accommodated into the oval-shape pocket of the ERH-2 dimer (Fig. 4b, c). The helix of TOST-1 (aa 39–48) interacts with both ERH-2 protomers, ERH-2$^{A}$ and ERH-2$^{B}$, via hydrophobic and electrostatic interactions (Fig. 4d). Leu39, Phe43, and Leu46 of TOST-1 make extensive

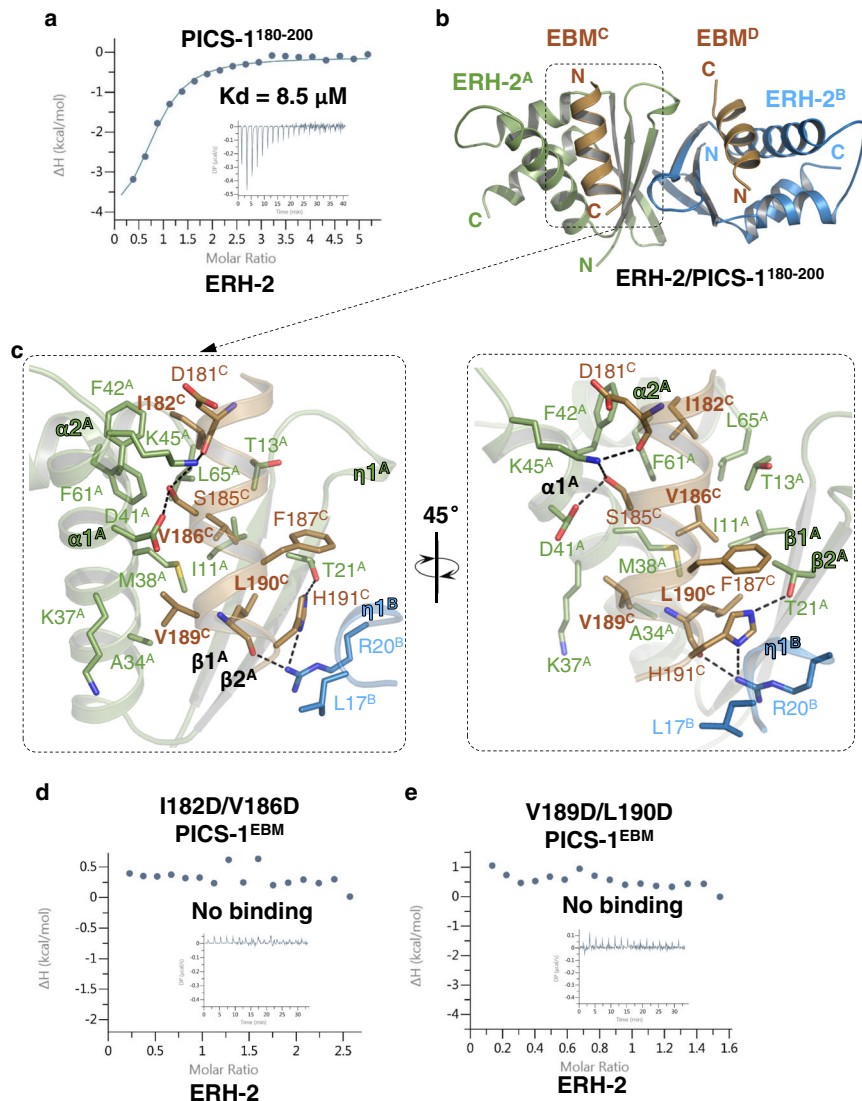

**Fig. 3 The structure of ERH-2 dimer bound with EBM. a** ITC binding curves for EBM with ERH-2. **b** Overall structure of the ERH-2 dimer with two EBM molecules. The two protomers in ERH-2 dimer, ERH-2$^A$ and ERH-2$^B$, are shown in green and blue cartoon, respectively. Two EBM molecules, EBM$^C$ and EBM$^D$, that bind to the ERH-2 dimer symmetrically, are shown in brown cartoon. **c** Detailed interactions between EBM$^C$ and ERH-2 dimer. Residues involved in ERH-2/EBM interaction are shown in sticks. Hydrogen bonds are shown in black dashes. **d**, **e** ITC curves for EBM mutants with ERH-2.

hydrophobic interactions with Val8$^A$, Trp22$^A$, Phe73$^A$, Tyr80$^A$, Leu10$^B$, and Met71$^B$. The main chain amides of Leu39 and Asn40 form are hydrogen bonded to the carboxyl groups of Asp67$^B$; The side chain carboxyl group of TOST-1 Asn40 is hydrogen bonded to the main chain amide group of Arg85$^B$; The guanidino group of Arg42 forms cation-π and salt bridge with the side chains of His6$^A$ and Glu26$^A$, respectively (Fig. 4d). In addition to the helix-mediated interactions, the Arg36 of TOST-1 forms salt bridge with Asp24$^A$, and the Thr51 of TOST-1 also makes additional hydrophobic contacts with Phe73$^A$ (Fig. 4d).

Then we superimposed the two ERH-2 complexes together after modeling the other TOST-1 molecule based on symmetry. In the overlaid structures, two EBM and two TOST-1 occupy different surfaces of ERH-2 dimer (Supplementary Fig. 7), which is consistent with our structural analysis that ERH-2 interact with EBM and TOST-1 via different residues (Supplementary Fig. 5a). Collectively, the two solved ERH-2 complexes imply that ERH-2 homodimer acts as a scaffold to bridge two PICS subunits in vitro.

Next we mutated the residues at the ERH-2/TOST-1 interface to evaluate their roles in complex formation. By using ITC

binding assay, we found that D67A of ERH-2, R42C and L39A/F43A of TOST-1, abolished the interaction between TOST-1 and ERH-2 (Fig. 4e–g), whereas the ERH-2 D67A binds to EBM in a comparable affinity with the wild type (K$_d$s: 4.3 μM vs. 8.5 μM) (Supplementary Table 1). Taken together, the binding data and mutagenesis experiments suggest that the PICS-1(PID-3), and TOST-1 or PID-1, are capable of binding to different surfaces of dimerized ERH-2 simultaneously.

Furthermore, we mixed ERH-2 and TOFU-6$^{RRM}$ with almost full-length PICS-1(PICS-1$^{5-282}$), PICS-1$^{5-200}$(PICS-1$^{\Delta RRM}$), and PICS-1$^{5-282}$ with EBM deleted (PICS-1$^{\Delta EBM}$), respectively, and purified them by gel filtration (Supplementary Fig. 8a). Gel filtration assay showed that PICS-1$^{5-282}$ interacts with ERH-2 and TOFU-6$^{RRM}$ to form a complex of ~100 kDa (Supplementary Fig. 8b), suggesting that the complex contains two copies of each subunit. In contrast, PICS-1$^{\Delta RRM}$ binds to ERH-2, but not TOFU-6$^{RRM}$, whereas PICS-1$^{\Delta EBM}$ binds to TOFU-6$^{RRM}$, but not ERH-2 (Supplementary Fig. 8c-d). Given that the ERH-2 homodimer binds to two PICS-1(PID-3) and two TOST-1 molecules simultaneously and PICS-1(PID-3) also binds to

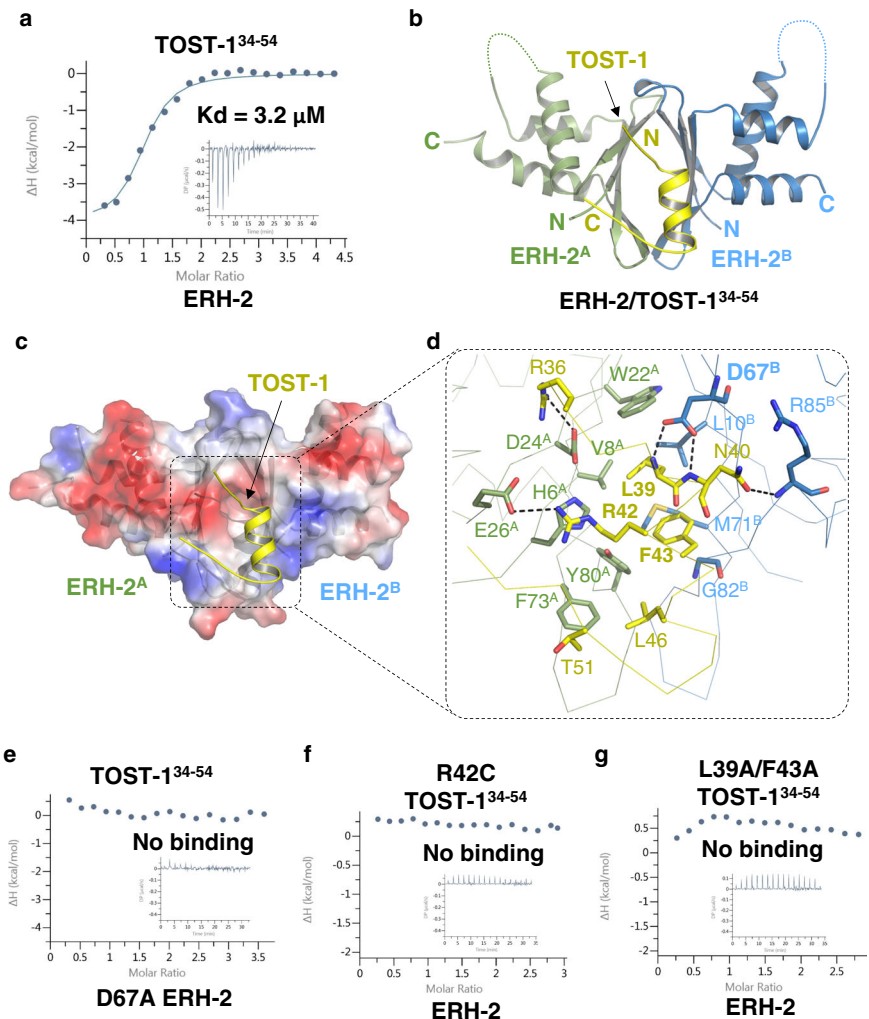

**Fig. 4 The structure of ERH-2 dimer with TOST-1$^{34-54}$. a** ITC binding curve for TOST-1$^{34-54}$ with ERH-2. **b** Overall structure of the ERH-2 dimer with one TOST-1$^{34-54}$, with the other TOST-1$^{34-54}$ blocked by crystal packing. ERH-2 dimer are shown in the same way as in Fig. 3b, and invisible loop regions of ERH-2 dimer are indicated by dashes. **c** Electrostatic surface of ERH-2 dimer bound to TOST-1$^{34-54}$. **d** Detailed interactions between TOST-1$^{34-54}$ and ERH-2 dimer. Residues involved in ERH-2/TOST-1 interaction are shown in sticks. Hydrogen bonds are shown in black dashes. **e** ITC binding curve for TOST-1$^{34-54}$ with the ERH-2 D67A. **f, g** ITC binding curves for mutants of TOST-1$^{34-54}$ with wild type ERH-2.

ERH-2 and TOFU-6$^{RRM}$ via EBM and RRM, respectively (Supplementary Fig. 7), our data further support that PICS complex is an octamer consisting of two copies of each subunit.

**Impact of interaction network on subcellular localizations.** We further examined how the in vivo localizations of PICS subunits are affected by the interaction network that has been uncovered by in vitro experiments. Consistent with our previous work, TOFU-6 is localized in the cytosol and the perinuclear granules. The enrichment of TOFU-6 in the perinuclear granules depends on PICS-1(PID-3) and other PICS subunits (Fig. 5a)[39]. TOFU-6 D26A/W27A, which severely impairs the TOFU-6/PICS-1 inter-action, is still in cytoplasm, but does not associate with peri-nuclear granules in germline cells (Fig. 5a). In contrast, TOFU-6 wild type and D26A/W27A mutant localize in the cytosol without association with granule in oocytes and embryos (Supplementary Fig. 9). PICS-1(PID-3) is localized in cytosol in germline cells, oocytes and embryos (Fig. 5b and Supplementary Fig. 10a), and associates with the perinuclear granules in germline cells (Fig. 5b). The PICS-1(PID-3) F217E mutant, which disrupts the homo-dimerization, failed to associate with perinuclear granules (Fig. 5c). K246A/F247A and Y250A/Q251A of PICS-1(PID-3)

that remarkably weakened the TOFU-6/PICS-1 interaction, failed to associate with perinuclear granules but a large fraction of the proteins localized inside the nuclear envelope and accumulated in the nucleoplasm in germline cells (Fig. 5c), as evidenced by using lamin protein LMN-1::mCherry as the nuclear envelope marker (Supplementary Fig. 10b). In contrast, the three PICS-1(PID-3) mutants exhibited the same localization as the wild type protein in oocytes and embryos (Supplementary Fig. 10c). For cells with *tofu-6* mutants, PICS-1(PID-3) failed to accumulate at the peri-nuclear granules in the germline, but a large fraction of the protein changed the localization from the cytoplasm to nucleo-plasm in germline and embryos (Fig. 5b and Supplementary Fig. 11a). Taken together, the localization of PICS-1(PID-3) in the cytosol depends on TOFU-6, and the TOFU-6/PICS-1 interaction is critical for both proteins to associate with perinuclear granules.

Then we sought to examine how the PICS-1/ERH-2 interaction affects the localizations of both proteins. In germline cells, the two PICS-1(PID-3) mutants eliminating the interaction between PICS-1(PID-3) and ERH-2, I182D/V186D, and V189D/L190D, form more and bigger perinuclear granules in germline cytoplasm in *pics-1* mutant background, similar to that of wild-type PICS-1(PID-3) in *erh-2* mutant background (Fig. 5b-c), indicating that

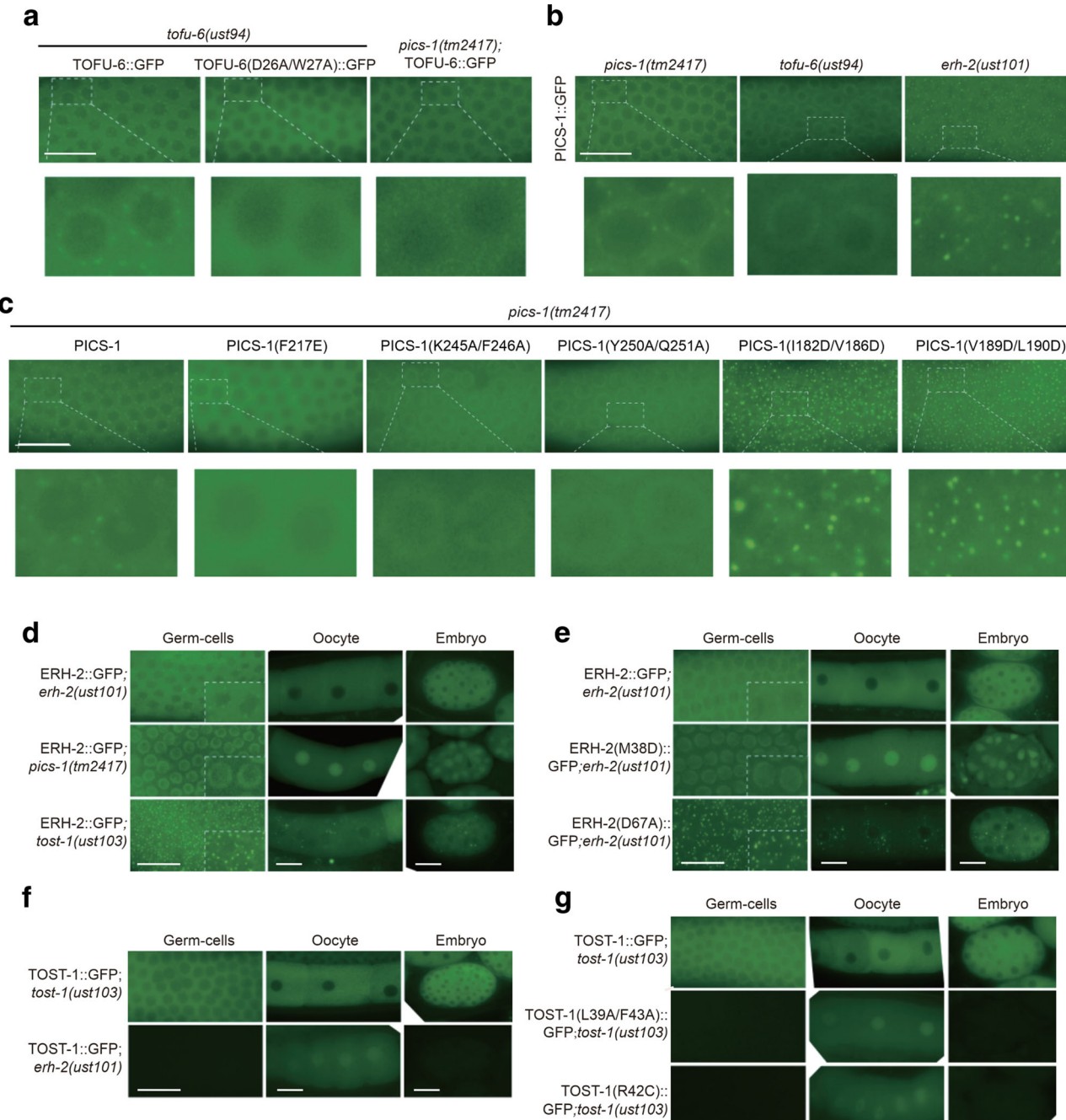

**Fig. 5 PICS promoting 21U RNAs biogenesis are dependent of the interactions of PICS subunits. a** Images of adult germline cells expressing GFP tagged-TOFU-6 or TOFU-6 D26A/W27A in indicated mutant backgrounds. TOFU-6 D26A/W27A impairs the interaction between TOFU-6 and PICS-1(PID-3). **b** Images of adult germline cells expressing GFP tagged- PICS-1(PID-3) in indicated mutant backgrounds. **c** Images of adult germline cells expressing GFP tagged wild-type or amino acids- substituted PICS-1(PID-3) in *pics-1 (tm2417)* background. K245A/F246A and Y250A/Q251A of PICS-1(PID-3) weaken the interaction with TOFU-6 . F217E disrupts the homodimerization of PICS-1(PID-3). I182D/V186D and V189D/L190D of PICS-1(PID-3) disrupt the interaction with ERH-2. **d** Images of adult germline cells, oocytes and embryos expressing GFP tagged ERH-2 in indicated mutant backgrounds. **e** Images of adult germline cells, oocytes and embryos expressing GFP tagged wild-type or amino acids- substituted ERH-2 in *erh-2* null mutant background. ERH-2 M38D disrupts the interaction with PICS-1(PID-3) . ERH-2 D67A disrupts the interaction with TOST-1. **f** Images of adult germline cells, oocytes and embryos expressing GFP tagged TOST-1 in indicated mutant backgrounds. **g** Images of adult germline cells, oocytes and embryos expressing GFP tagged wild-type or amino acids- substituted TOST-1 in *tost-1* null mutant background. L39A/F43A and R42C of TOST-1 disrupt the interaction with ERH-2. Scale bars, 20 μm. The worms were harvested once, while the imaging experiments were triplicated.

the interaction between PICS-1(PID-3) and ERH-2 plays an important role in controlling the level of PICS-1(PID-3) in granules. ERH-2 localizes in the cytosol and associates with perinuclear granules in germline cells, and resides in the cytoplasm in oocytes and embryos (Fig. 5d and supplementary Fig. 11b). The ERH-2 M38D mutant that also abolished the PICS-1/ERH-2 interaction, is located in the nucleus in germline cells, oocytes and embryos (Supplementary Table 1, Fig. 5e), similar to that observed for ERH-2 when *pics-1(pid-3)* was mutated (Fig. 5d). *tofu-6* mutations that impair the TOFU-6/PICS-1 interaction, also resulted in a nuclear accumulation of ERH-2 in embryos (Supplementary Fig. 11b), demonstrating the essential role of TOFU-6 in directing PICS-1(PID-3) and ERH-2 into perinuclear granules.

Next we studied the impact of ERH-2/TOST-1 interaction on their subcellular localizations. In germline cells, ERH-2 forms more and bigger granules in *tost-1* mutants than in wild type animals (Fig. 5d). D67A of ERH-2 that disrupts the binding of ERH-2 to TOST-1, forms more granules in the cytoplasm (Fig. 5e), suggesting the role of TOST-1 in controlling the assembly of ERH-2 associated perinuclear granules. TOST-1 localized at the cytoplasm of germline cells, oocytes, and embryos in wild type animals (Fig. 5f). In *erh-2* mutant, TOST-1 is not expressed in the germline cells and embryos, but accumulated in the nucleus in oocytes (Fig. 5f). The two TOST-1 mutants that disrupt the ERH-2/TOST-1 interaction, L39A/F43A and R42C, are not expressed in germline cells and embryos, but accumulate in the nucleus in oocytes (Fig. 5g). We further constructed GFP tagged PID-1(R61C) transgene, a mutation abolishing the ERH-2/PID-1 interaction and leading to reduced piRNA levels [38,40]. Consistent with previous Western blot assay [38,40], the expression of PID-1(R61C)::GFP is low and similar to that of TOST-1(R42C)::GFP in germ-cells (Fig. 5g and Supplementary Fig. 12). In summary, genetic experiments indicate that the localizations of ERH-2 and TOST-1 are dependent on each other.

**Intact PICS complex is required for chromosome segregation and cell division.** Consistent with our previous work that the TOST-1-containing PICS complex regulates embryonic chromosome segregation and cell division [39], null mutants of PICS subunits, such as *tofu-6(ust94)*, *erh-2(ust101)*, *pics-1(tm2417)*, and *tost-1(ust103)*, all lead to delay and nondisjunction in chromosome segregation during cell division (Fig. 6a-d). In contrast with the null mutants of PICS subunits, none of the three PICS-1(PID-3) mutants impairing TOFU-6/PICS-1 interaction (K245A/K246A, Y250A/Q251A, and F217E), has any impact on chromosome segregation and cell division (Fig. 6b), neither does the TOFU-6 mutant *ust173* (Fig. 6a), suggesting that full-length TOFU-6 but not its PICS-1-binding domain , is essential for chromosome segregation and cell division. It is likely that other region of TOFU-6, such as the Tudor domain or the C-terminal region, mediates chromosome segregation and cell division.

Next we examined whether interactions between PICS-1(PID-3), ERH-2 and TOST-1 are required for the maintenance of normal chromosome segregation during cell cycle. We found that the three mutants disrupting the ERH-2/PICS-1 interaction, including I182D/V186D and V189D/L190D of PICS-1(PID-3), and M38D of ERH-2, all display abnormal chromosome segregation and cell division (Fig. 6b-c). In addition, those mutants abolishing the ERH-2/TOST-1 interaction, including D67A of ERH-2, L39A/F43A and R42C of TOST-1, all demonstrated delay and nondisjunction in chromosome segregation and cell division (Fig. 6c-d). Consistently, above mutants that caused defects in chromosome segregation and cell division, also

exhibited maternal effect embryonic lethality and resulted in sterility of the animals (Supplementary Fig. 13). Therefore, our data indicate that the PICS-1-ERH-2-TOST-1 axis is critical for chromosome segregation, cell division, and fertility.

Given the known role of PICS in piRNA biogenesis, the mis-localizations of PICS subunits also prompt us to investigate whether the mutants associated with aberrant perinuclear granule localization also have impact on piRNA biogenesis in vivo. We examined the levels of typical piRNAs by quantitative real-time PCR and found that those missense mutants, as well as the null mutants, all lead to impaired piRNA production in vivo (Fig. 7a). Collectively, intact interaction network within PICS is essential not only for the association with perinuclear granules, but also for piRNA biogenesis in vivo.

## Discussion
piRNAs, termed for their association with PIWI proteins, play an important role in defending against non-self nucleic acids. In *C. elegans*, piRNAs are abundant in germline and necessary for fertility. Recently, a variety of protein complexes have been identified to be involved in piRNA biogenesis, including transcription, export, and maturation. PICS complex consists of four core subunits and is involved into the regulation of piRNA biogenesis and cell division via two mutually exclusive factors, PID-1 and TOST-1, respectively [38,39].

Here we present three structures for PICS subunits, including those of TOFU-6^RRM/PICS-1^RRM, ERH-2/EBM and ERH-2/TOST-1, to uncover the mechanism underlying the PICS formation. Complemented by biochemical and mutagenesis experiments, our structural biology data support that PICS is an octamer composed of two copies of each subunit, and unveil the assembly interface within PICS complex. The interactions between PICS subunits are critical for the proper localization of PICS in vivo, suggesting the localization of PICS units are dependent on each other. Consistent with previous work [38,39], PICS-1(PID-3) and TOFU-6 interact with each other via RRM domains, which were previously identified as the RNA binding modules. PICS-1 binds to TOFU-6 and ERH-2 via RRM and EBM, respectively. ERH-2 and PICS-1(PID-3) are both homodimers within the PICS complex and the two homodimers interact with each other in a 1:1 ratio to form a homogenous tetramer rather than high-order polymers or aggregates, which constitutes the basis of PICS octamer (Fig. 7b).

**Functional implication of ERH-2 and its eukaryotic orthologs.** ERH-2 is the only conserved subunit in PICS complex among eukaryotic species and the solved structures of *C. elegans* ERH-2 confirmed that a shared surface of ERH-2 dimer mediates the binding of TOST-1 or PID-1, allowing PICS complex to make a choice in being involved into piRNA biogenesis or cell division. The overall structure of ERH-2, as well as the dimer interface, are also conserved from yeast to human (Supplementary Fig. 5a). Recently, two eukaryotic ERH-2 complexes were solved, including the structures of human ERH bound with a fragment of DGCR8, and of *Schizosaccharomyces pombe* Erh1 with Mmi1, which prompts us to compare the structures of *C. elegans* ERH-2 with its orthologs [54,55].

In all structures, ERH forms homodimer and binds to ligands in a 2:2 ratio. In the overlaid structures, we found that although ERH orthologs utilize similar surface to bind their ligands, the ligands adopt different conformations upon binding to different ERH orthologs (Supplementary Fig. 5b). Despite that the DGCR8 fragment overlaps with EBM, it exhibits extended conformation whereas the EBM adopts a helix [55]. It is possible that some unknown protein might interact with human ERH via the TOST-

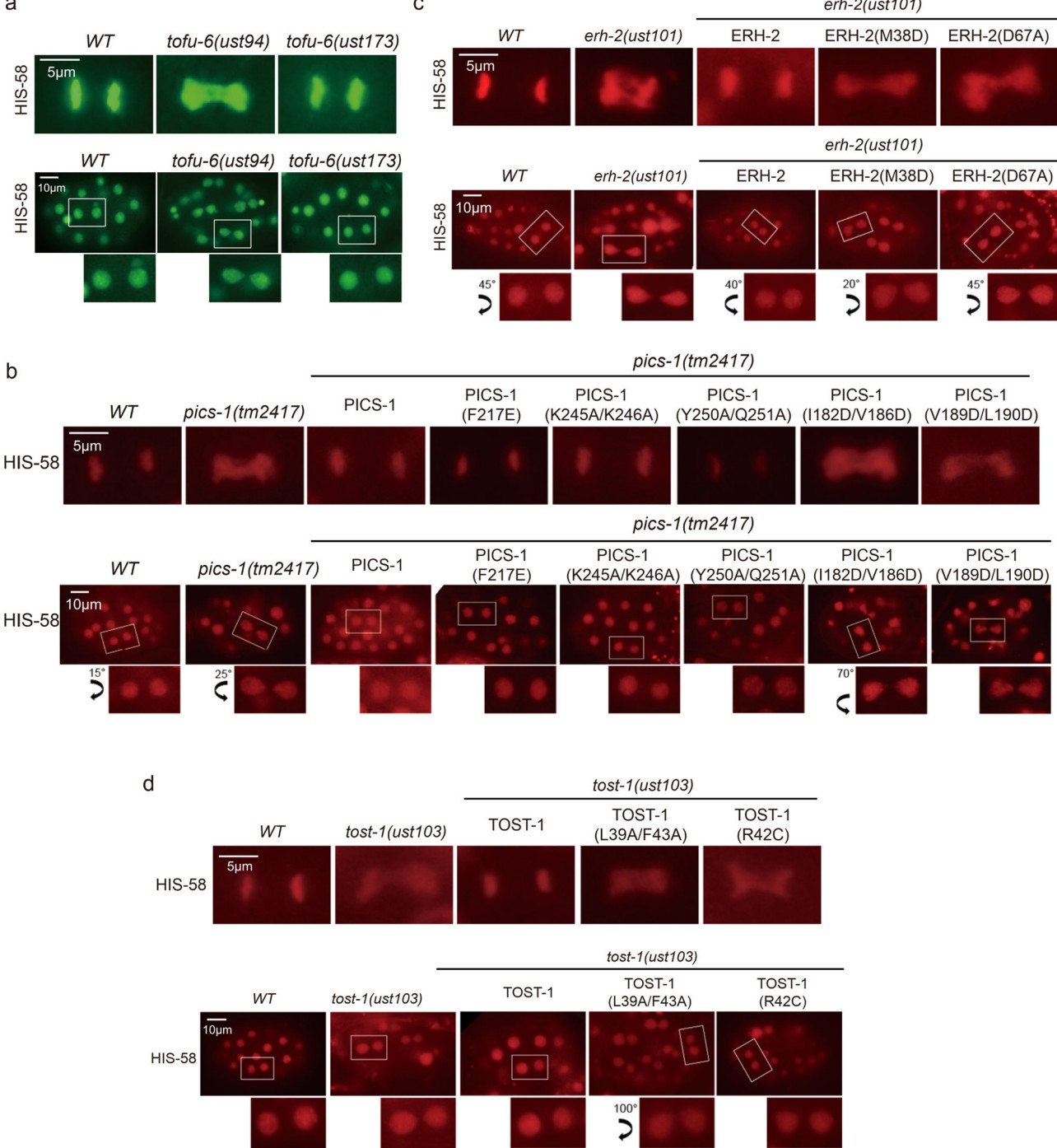

**Fig. 6 Disruption of the PICS-1/ERH-2 or ERH-2 & TOST-1 interaction, but not the TOFU-6/PICS-1 interaction, leads to defective chromosome segregation and cell division. a–d** Images of GFP or mCherry tagged HIS-58 at the meta-anaphase in embryos in indicated backgrounds. The worms were harvested once, while the imaging experiments were triplicated.

1 binding site (Supplementary Fig. 5b). The N-terminal and C-terminal fragments of *Schizosaccharomyces pombe* Mmi1 overlaps with TOST-1 and EBM, respectively. While the N-terminal fragment of Mmi1 adopts a distorted helix, TOST-1 demonstrates a regular helix structure with two extended fragments at both ends. The C-terminus of the Mmi1 polypeptide exhibits an extended conformation, different from that of EBM (Supplementary Fig. 5)[54]. It is noteworthy that neither of the *C. elegans* ERH-2 binding ligands is conserved in *homo sapiens* or *Schizosaccharomyces pombe*, and vice versa. Despite the versatile ligand binding modes observed in eukaryotic ERH complexes,

ERH dimer likely maintains the conserved role as the scaffold through keeping the surfaces for dimerization and for ligand binding during evolution.

Although the binding ligands for ERH-2 and its orthologs are not conserved, *Schizosaccharomyces pombe* Erh1 and human ERH were also reported to be associated with cell division[43]. Specifically, Erh1-Mmi1 plays an important role in degrading specific transcripts during meiosis[44], whereas mammalian ERH is indispensable for chromosome segregation during mitosis because of its key role in regulating the expression of CENP-E, a centromere-associated kinesin-like motor protein[45,46]. The

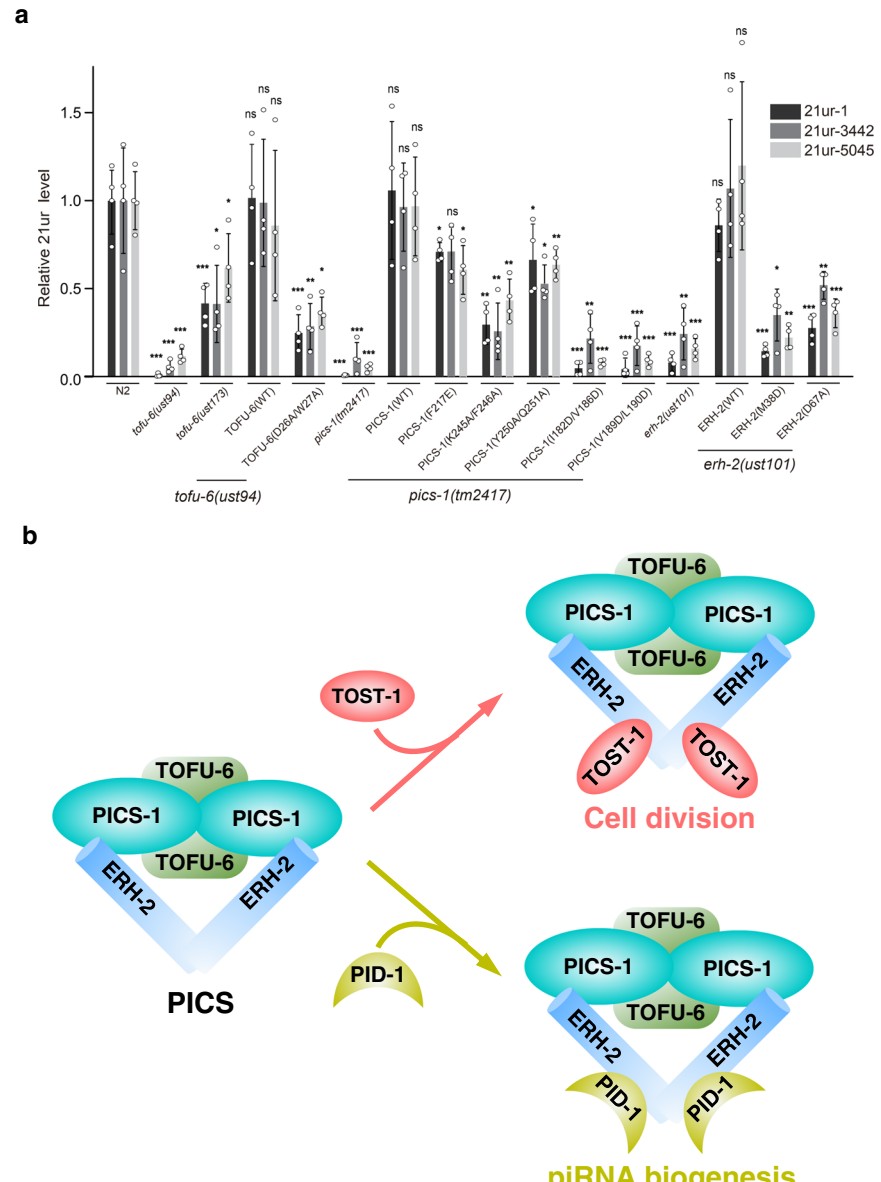

**Fig. 7 piRNA biogenesis depends on the interactions between PICS subunits. a** 21U-RNA levels are detected by quantitative RT-PCR with RNA levels normalized to total reverse transcribed RNAs. Mean ± SD; ***$p < 0.001$, **$p < 0.01$, *$p < 0.05$; ns, not significant; $n = 4$ replicates; two-tailed paired t-tests. **b** A working model for PICS complex. PICS complex consists of four core subunits, and functions in chromosome segregation and piRNA biogenesis via two mutually exclusive factors, TOST-1 and PID-1, respectively.

diverse functions of mammalian ERH are attributed to its capacity of binding with different ligands, such as DGCR8, POLDIP3, and CIZ1, *etc*[44]. Similarly, the role of *C. elegans* ERH-2 in mitosis could also be relevant to its potential nuclear function in gene regulation via binding to TOST-1 and other unknown factors. Furthermore, through binding to TOST-1 and PID-1 in a mutual manner, ERH-2 acts as an anchor protein not only in conferring TOST-1 the ability to antagonize piRNA biogenesis, but also in establishing a fine-tuned spatial and temporal crosstalk between the two key cellular events.

**The role of PICS in cell division.** PICS complex has dual roles in piRNA biogenesis and chromosome segregation. The correct localizations of PICS subunit are interdependent. Specifically, the cytoplasmic localizations of the other PICS subunits are dependent on the interaction between TOFU-6 and PICS-1(PID-3), suggesting the role of TOFU-6 in balancing the levels of PICS

subunits between cytoplasm and nucleus. Intriguingly, the TOFU-6/PICS-1 interaction is important for piRNA biogenesis, but not required for chromosome segregation, although TOFU-6 is essential for chromosome segregation, one explanation could be that TOFU-6 might mediate chromosome segregation via its N-terminal MID domain or C-terminal IFE-3 binding motif (Fig. 1a). In contrast, the interactions between PICS-1(PID-3), ERH-2 and TOST-1 are essential in chromosome segregation during cell division. Therefore, PICS mediates cell division via the PICS-1/ERH-2/TOST-1 axis, in which ERH-2 dimer acts as the scaffold to tether PICS-1(PID-3) and TOST-1. In addition, TOST-1 is cytoplasmic but becomes primarily nuclear if *erh-2* is mutant, and this is also true for TOST-1 mutants that disrupt the TOST-1/ERH-2 interaction. It is likely that ERH-2/TOST-1 interaction is required for maintaining the ERH-2 level in the nucleus, which might be essential for cell division or spliced leader maturation.

**The role of PICS in piRNA biogenesis.** As for the role of PICS complex in piRNA biogenesis, one key question to be answered is how PICS complex is targeted to piRNA precursors. There are two possibilities, one is that IFE-3, known to bind the 5' cap of piRNA, might recruit PICS via the interaction with the IBM of TOFU-6. The other is that PICS might directly bind to piRNA precursors via the PICS-1$^{RRM}$-TOFU-6$^{RRM}$ tetramer. The positively charged surface of the tetramer suggests that it could be the potential binding site for piRNA ligand. Given the ultrahigh binding affinity between PICS-1$^{RRM}$ and TOFU-6$^{RRM}$ ($K_d$: 6.2 nM) (Fig. 1b), it is likely that the tetramer binds to piRNA precursor as a whole and the positively charged surfaces of PICS-1$^{RRM}$ and TOFU-6$^{RRM}$ might coordinate to bind to longer RNAs.

Previously, we and Ketting group have independently identified IFE-3 associated with TOFU-6, PICS-1(PID-3), ERH-2, PID-1 and TOST-1 from IP-MS, and found that recombinant IFE-3 directly binds to the IBM motif of TOFU-6[38,39]. IFE-3 acts as the binding protein for 5' capped piRNAs and is enriched in P-granules[38,39]. Also *ife-3* mutant leads to defects in piRNA biogenesis and maternal effect lethality[38]. However, more evidence imply that the functions of IFE-3 do not fully overlap with those of PICS complex. Firstly, IFE-3 but not PICS, binds directly to capped piRNA precursors[38]. Secondly, *tofu-6* mutant worms but not *ife-3* null mutant worms, have germline and eggs, suggesting that *ife-3* null mutant is more severe. Thirdly, IFE-3 accumulates in peri-nuclear granules in germ-cells and embryos, whereas TOFU-6 is localized in granules in germ-cells but not in embryos[38,39]. Fourthly, unlike the mutual dependence of PICS factors, the perinuclear localization of TOFU-6 is independent of *ife-3*[39]. Fifthly, although the role of IFE-3 in piRNA biogenesis was reported, its function in cell division is largely unknown.

Overall our structural biology work, complemented by biochemistry, cell biology and imaging experiments, not only unveils the mechanism underlying the roles of PICS-1(PID-3) and ERH-2 as scaffolds within the PICS complex, but also provides insights into understanding the dual functions of PICS through targeting one of the two mutually exclusive factors, PID-1 or PICS-1(PID-3). Therefore, the existence of PICS-like complex might reflect the requirement of higher eukaryotes for accurately mediating the balance between key cellular events, such as piRNA biogenesis and cell division.

## Methods

**Cloning, mutations, protein expression and purification.** Gene encoding full-length *C. elegans* TOFU-6, PICS-1(PID-3), TOST-1 and PID-1 were codon-optimized for expression in *Escherichia coli* and synthesized by Sangon Biotech. ERH-2 was amplified by PCR from the *C. elegans* cDNA library. TOFU-6$^{RRM}$(residues 2-92) and PICS-1$^{RRM}$(residues 201-282) were cloned into upstream and downstream multiple cloning sites of a modified pETDuet-1 (Novagen). ERH-2$^{1-103}$-(Gly-Ser-Ser)$_7$-PICS-1$^{180-200}$ and ERH-2$^{1-103}$-(Gly-Ser-Ser)$_6$-TOST-1$^{34-54}$ fusion clone were constructed by overlap extension PCR, and were cloned into pET28-MHL. EBM of PICS-1(PID-3) (residues 180-200) and a TOST-1 fragment (residues 34-54) were cloned into pET28-SUMO vector (Invitrogen™).

Recombinant proteins were overexpressed in *Escherichia coli* BL21(DE3). Cells were grown in LB medium at 37 °C until the optical density (OD$_{600}$) reached ~0.8. Protein expression was induced with 0.2 mM β-d-1-thiogalactopyranoside for 20 h at 16 °C. Cells were gathered by centrifuge at 3,600 g for 10 min at 4 °C, and pellets were resuspended in lysis buffer containing 20 mM Tris-HCl, pH 7.5, 400 mM NaCl.

For All recombinant proteins and protein complex were purified firstly by Ni-NTA (GE healthcare). Tobacco etch virus (TEV) protease was added to remove the N-terminal tag. After dialysis with lysis buffer overnight, the mixture was applied to another Ni-NTA resin to remove the protease and uncleaved proteins. TOFU-6/PICS-1 was further purified by HiLoad 16/600 Superdex 75 column (GE healthcare) in a buffer containing 20 mM Tris-HCl, pH8.5, 200 mM NaCl. Seleno-methionine (SeMet) labeled TOFU-6/PICS-1 was purified in the same way as native proteins, except that cells were cultured in M9 medium supplied with 50 mg/L of SeMet. ERH-2$^{1-103}$-(Gly-Ser-Ser)$_7$-PICS-1$^{180-200}$ and ERH-2$^{1-103}$-(Gly-Ser-Ser)$_6$-TOST-1$^{34-54}$ were further purified by HiLoad 16/600 Superdex 75 column (GE healthcare) in a buffer containing 20 mM MES, pH 6.5, 200 mM NaCl. PICS-1$^{5-282}$, PICS-1$^{5-200}$, and PICS-1$^{\Delta EBM}$ were cloned into pET28-MHL, expressed, and purified in the same way.

Site-specific mutations were performed using two reverse and complement primers containing the mutation codon. Primer sets used for mutations are listed in Supplementary Tables 2 and 3. The mutants of TOFU-6$^{RRM}$, PICS-1$^{RRM}$, and ERH-2 were cloned into the pET28-MHL vector, while the mutants of EBM and TOST-1 were cloned into the pET28-SUMO vector. All mutants were purified in the same way as wild-type proteins.

**Crystallization, data collection, and structure determination.** Protein crystals were grown at 18 °C by sitting drop vapor diffusion method. For crystallization, 1 μl of protein solution was mixed with 1 μl of crystallization buffer. TOFU-6/PICS-1 complex were concentrated to 10–15 mg ml$^{-1}$ in a buffer containing 20 mM Tris–HCl, pH 8.5, 200 mM NaCl; ERH-2$^{1-103}$-(Gly-Ser-Ser)$_7$-PICS-1$^{180-200}$ and ERH-2$^{1-103}$-(Gly-Ser-Ser)$_6$-TOST-1$^{34-54}$ were concentrated to 10–15 mg ml$^{-1}$ in a buffer containing 20 mM MES, pH 6.5, 200 mM NaCl. Crystals were obtained as follows:

(1) Native TOFU-6$^{RRM}$/PICS-1$^{RRM}$ complex was crystallized in a buffer containing 0.1 M Tris hydrochloride, pH 8.5, 2 M Ammonium sulfate.
(2) SeMet TOFU-6$^{RRM}$/PICS-1$^{RRM}$ complex was crystallized in a buffer containing 2% (v/v) 1,4-Dioxane, 0.1 M Tris hydrochloride, pH 8.0, 15% PEG 3350 and 0.01 M Nickel (ll) chloride hexahydrate.
(3) ERH-2$^{1-103}$-(Gly-Ser-Ser)$_7$-PICS-1$^{180-200}$ was crystallized in a buffer containing 15% (v/v) 2-propanol, 0.1 M sodium citrate tribasic dehydrate, pH 5.0 and 10% PEG 10000.
(4) ERH-2$^{1-103}$-(Gly-Ser-Ser)$_6$-TOST-1$^{34-54}$ was crystallized in a buffer containing 0.1 M Bis-tris, pH 6.2, 20% tascimate (pH 6.0).

Before flash-freezing crystals in liquid nitrogen, all crystals were soaked in a cryo-protectant consisting of 85% reservoir solution plus 15% glycerol. The diffraction data were collected at 0.9785 or 0.9792 Å on beamline BL18U1 and BL19U1[47] at Shanghai Synchrotron Facility. The statistic details about data collection are summarized in Table 1. Data sets were processed using HKL2000/3000[48,49] or XDS[50]. The initial model of SetMet TOFU-6/PICS-1 complex was solved by CRANK2[51], built manually by COOT[52], and refined by Phenix[53]. Structure of native TOFU-6/PICS-1 was solved by Phenix[53] with the SeMet structure as the search model for molecular replacement. ERH-2$^{1-103}$-(Gly-Ser-Ser)$_7$-PICS-1$^{180-200}$ and ERH-2$^{1-103}$-(Gly-Ser-Ser)$_6$-Tost-1$^{34-54}$ structures were solved by molecular replacement with human ERH protein (PDB ID: 1WZ7) as the search model and were further refined by Phenix[53].

After refinement, the structure of TOFU-6$^{RRM}$ (L73M)-PICS-1$^{RRM}$ (I233M) (SeMet) was refined to 1.95 Å (favored 98.4%, allowed 1.6%); the structure of TOFU-6$^{RRM}$-PICS-1$^{RRM}$ (Native) was refined to 2.68 Å (favored 98.1%, allowed 1.9%); the structure of ERH-2$^{1-103}$-(GSS)$_7$-PICS-1$^{180-200}$ was refined to 2.40 Å (favored 97.1%, allowed 2.9%); the structure of ERH-2$^{1-103}$-(GSS)$_6$-TOST-1$^{34-54}$ was refined to 2.20 Å (favored 99.0%, allowed 1.0%). Stereo images of representative 2|Fo|−|Fc| maps for crystal structures contoured at 1.0 σ were shown in Supplementary Fig. 14.

**Isothermal titration calorimetry (ITC).** ITC binding assays were carried out on a MicroCal ITC200 calorimeter (GE Healthcare) at 25 °C in a buffer containing 20 mM Tris, pH 8.5 , 200 mM NaCl or 20 mM MES, pH 6.5, 200 mM NaCl. Before ITC experiments, all purified proteins were dialyzed against ITC buffer. ITC experiments were performed by titrating 2 μl of proteins or peptides in the syringe (1–2 mM) into the cell containing 30–50 μM proteins, with a spacing time of 120 s and a reference power of 5 μCal s$^{-1}$. Control experiments were performed by titrating proteins or peptides (1–2 mM) into the ITC buffer, and were subtracted during analysis, respectively. Binding isotherms were plotted, analyzed and fitted based on one-site binding model by MicroCal PEAQ-ITC Analysis software (Malvern Panalytical) after subtraction of respective controls. The dissociation constants (Kd) and peptide sequences used for ITC experiments are summarized in Supplementary Table 1, and representative ITC binding curves were included in Source Data file.

**Gel filtration assay.** For gel filtration assay, the wide type and mutant (F217E) of PICS-1$^{RRM}$ were purified in by HiLoad 16/600 Superdex 75 column (GE healthcare) in a buffer containing 20 mM Tris hydrochloride, pH 8.5, and 200 mM NaCl. Different PICS subcomplexes, including TOFU-6$^{RRM}$/PICS-1$^{5-282}$/ERH-2, PICS1$^{\Delta RRM}$/ERH-2, and TOFU-6$^{RRM}$/PICS1$^{\Delta EBM}$, were purified by Superose 6 Increase 10/300 GL (GE healthcare) in a buffer containing 20 mM Tris hydrochloride, pH 8.5, and 200 mM NaCl. The scanned images for the SDS-PAGE gels of protein samples after gel filtration were included in Source Data file.

**Construction of transgenic mutant strains.** For wild-type and amino acids substituted GFP tagged transgenes, endogenous promoter sequences, ORFs or amino acids substituted ORFs, coding sequence for gfp::3xflag, 3′ UTR, and a linker sequence (GGAGGTGGAGGTGGAGCT) (inserted between the ORFs and gfp::3xflag) were fused and cloned into PCFJ151 vectors using a ClonExpress

MultiS One Step Cloning Kit (Vazyme C113-02, Nanjing). These transgenes were integrated into the *C. elegans'* chromosome II or V by MosSCI technology.

For *tofu-6(ust173)* mutant, sgRNAs closed to the substituted bases and the repair template were co-injected into N2 animals with PDD162 (50 ng/µL), pcfj90 (20 ng/µL), 10xTE buffer, and DNA ladder (500 ng/µL). The repair template that contains homologous arms of 1000 bp to 1500 bp and substituted base pairings were cloned into a vector using the ClonExpress MultiS One Step Cloning Kit. Two or three days later, F1 worms expressing the pcfj90 marker were isolated. After growing at 20 °C for another three days, the animals were screened for base pairing substitution by PCR and restriction enzyme reaction. Strains used in this work are listed in Supplementary Table 4.

**Microscope and images**. Images were collected on Leica DM4B microscope.

**RNA isolation**. Synchronized young adult worms were collected and sonicated in the sonication buffer containing 20 mM Tris hydrochloride, pH 7.5, 200 mM NaCl, 2.5 mM $MgCl_2$, and 0.5% NP40. The eluates were incubated with TRIzol reagent and then precipitated with isopropanol. The precipitated nucleic acids were treated with DNase I, re-extracted with TRIzol and then precipitated with isopropanol again. The concentrations of RNA were measured using Nano Drop.

**RT-qPCR**. RNA was isolated from the indicated animals and subjected to DNase I digestion (Thermo Fisher). cDNA was generated from the RNA using a *GoScript™ Reverse Transcription System* (Promega) according to the vendor's protocol. qPCR was performed using a MyIQ2 real-time PCR system (Bio-Rad) with AceQ SYBR Green Master mix (Vazyme). Total reverse transcribed RNA levels were used as an internal control for sample normalization.

**Brood size**. L3 worms were placed individually into fresh NGM plates, and the progeny numbers were scored.

**Statistics**. Bar graphs with error bars represented the mean and SD. All of the experiments were conducted with independent *C. elegans* animals for the indicated N replicates. Statistical analysis was performed with two-tailed Student's t-tests or unpaired Wilcoxon tests. Student's t-test p value threshold was set to 0.05.

**Reporting summary**. Further information on research design is available in the Nature Research Reporting Summary linked to this article.

## Data availability

The data supporting the findings of this study are available from the corresponding authors upon reasonable request. The crystal structures of SeMet TOFU-6$^{RRM}$(L73M)/PICS-1$^{RRM}$(I233M), native TOFU-6$^{RRM}$/PICS-1$^{RRM}$, ERH-2/EBM, and ERH-2/TOST-1 have been deposited in the Protein Data Bank (PDB) under accession codes 7D1L, 7D2Y, 7EJS, and 7EJO, respectively. Source data are provided with this paper.

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

## Acknowledgements
We thank the staffs from BL17B/BL18U1/BL19U1/BL19U2/BL01B beamline of the National Facility for Protein Science in Shanghai (NFPS) at Shanghai Synchrotron Radiation Facility, for assistance during data collection. This work is supported by the National Natural Science Foundation of China Grants (92053107), the National Key R&D Program of China (2019YFA0802600, 2018YFC1004500, and 2017YFA0102900). This work is also supported by the "Strategic Priority Research Program" of the Chinese Academy of Sciences (Grant No. XDB19000000, XDB39010600) and another National Natural Science Foundation of China Grants (31770806). C.X. is supported by "the Fundamental Research Funds for the Central Universities" and "the Thousand Young Talent program".

## Author contributions
C.X. and S.G. conceived the project. X.W. and S.L. performed structural biology and biochemical experiments with assistance from Z.Z., X.T. and J.Z. C.Z. and X.F. performed in vivo experiments. C.X. and S.G. wrote the manuscript with input from all authors. C.X. and S.G. supervised the project.

## Competing interests
The authors declare no competing interests.
