## [Peer Review File · Nature Communications]

Molecular basis for PICS-mediated piRNA biogenesis and cell divisionREVIEWER COMMENTS

Reviewer #1 (Remarks to the Author):

Wang et al., revealed the molecular basis for PICS complex assembly by using combined structural and biochemical methods. They also identified disruption of these interactions in vivo will lead to mislocalization of PICS subunits, piRNAs decreasing, or abnormal chromosome segregation in *C. elegans*. Overall, the data are of solid quality. However, there are some concerns which should be addressed before considering publication.

1. One of the main defects is that there are many misspellings in the text and in the Figures, making it difficult to read the article smoothly. The labelling in the Figures should be improved and be consistent with the text. For examples, Fig 3b (change PICS-1 to EBM as stated in the text) and so on. I also suggest to label Fig 1a with numbers to indicate the domain boundaries clearly as seen in the ITC tables.
2. Another shortcoming is the level of signal to noise is really low for the fluorescent images, making it hard to distinguish proteins localizations.
3. The article has no information about the PID-1 structure, it just showed PID-1 and TOST-1 mutually exclude each other from the PICS complex by ITC. My understanding is this article mainly focus on the TOST-1 containing complex. So the authors should focus on the chromosome segregation function instead of piRNA biogenesis. Regarding the impact of interaction network on subcellular localizations, I suggest the authors separate the piRNA biogenesis function with the chromosome segregation, instead of mixing them together.
4. Does the author have evidence to indicate that ERH-2 binding with PICS-1 will not affect the binding between PICS-1 with TOFU-6 using only RRM+EBM domain?
5. The authors should discriminate PIWI (subfamily name) and Piwi (protein name) throughout the text.
6. Author using a buffer containing 200mM NaCl for ITC analysis, is there any specific reason to use this high salt buffer?

Reviewer #2 (Remarks to the Author):

This manuscript provides the structural basis for molecular interactions in PICS complex composed of TOFU-6, PICS-1, ERH, and either of PID-1 or TOST-1, which is crucial for piRNA biogenesis and chromosome segregation/cell division, respectively. The authors solved the high-resolution crystal structures of the complexes of PICS-1(RRM) and TOFU-6 (RRM), of ERH-2 and PICS-1 fragment, and of TOST-1 fragment and ERF-2, and the atomic interactions are confirmed by ITC experiments of mutants. Moreover, they analyzed piRNA biogenesis and localization/cell division of the structure-designed mutants. While the structure determinations are solid, and the biochemical and cell biological results have impact to some extent, this reviewer has some concerns before acceptance of this paper.

1. As this manuscript does not provide any results of RNA-binding and DNA(chromatin) binding results in the structural viewpoint, this reviewer feels fluctuation that the structural results and the functional analyses seem separated. Therefore, this reviewer would like to ask the authors to provide some structural data of interaction between PICS complex and nucleic acids.
2. The authors presented the separate complex structures, but they describe that they could purify the complex consisting of ERH-2-(GSS)6-TOST-1. PICS-1(EBM-RRM), and TOFU-6(RRM) in gel filtration. Why did not they solve the complex structure by X-ray crystallography or Cryo-EM single particle analysis ?
3. The authors should pay more attention to prepare the manuscript. The followings are their mistakes:

- 1) In the first line of the introduction section, paly should be play.
- 2) In the second line of page 6, "homodimer" is duplicated.
- 3) In line 11 on the same page, they should add "and the side chain of Arg227B" just after Met213A.

Reviewer #3 (Remarks to the Author):

The authors and an independent group have previously identified the PICS complex, also known as PETISCO, and both previous papers arrived at remarkably consistent conclusions: that multiple components of PICS play roles in piRNA biogenesis, which is not essential for development or fertility, and in a distinct process that is essential for embryo viability. The previous studies identified two 'adaptor' proteins that mediate these functions - PID-1, which promotes piRNA biogenesis in conjunction with the 5' cap-binding factor IFE-3, and TOST-1, which inhibits piRNA function and is also essential for embryo development (chromosome segregation). It has previously been suggested that PID-1 and TOST-1 function in substrate recognition or processing. PID-1 and TOST-1 were previously shown to share a motif that binds to ERH-2 in a mutually exclusive manner. It was previously shown that the RRM domains of TOFU-6 and PICS-1 interact. The current manuscript presents detailed structural and functional analysis of several domains in PICS complex protein, *in vitro* and *in vivo*.

In the current study, the authors use elegant structural biology and biochemistry analysis to determine that the RRM domains of TOFU-6 and PICS-1 form homodimers that interact as a heterotetramer. Second, the authors show that ERH-2 has two distinct domains that interact with PID-1 or TOST-1, although these interactions are competitive and mutually exclusive. As with the RRM domains, the interactions between ERH-2 and PID-1 or TOST-1 are mutually exclusive. The major insight provided by the current manuscript is evidence for the stoichiometry of the PICS complex, where each subunit is present in two copies per PICS complex. The localization and function of PICS subunits where critical interacting amino acids are mutated is determined. This provides some insight into the subcellular distribution of PICS subunits. That said, there do not appear to be major new insights regarding either the piRNA maturation or the cell division functions of PICS subunits, apart from linking perinuclear granule localization to piRNA biogenesis, a concept that has already been well established in the field.

For example, in the Discussion, the authors point to previous ERH structures for *pombe* and human that are dimers, which is consistent with their model that PICS subunits are present in two copies per PICS complex. Beyond the presence of dimers of each subunit, it is not clear what this teaches us about PICS function. The combined structural biology and cell biology do not appear to provide significant insight into PICS function.

The authors solve crystal structures of domains of PICS proteins, which leads them to predict that the four proteins interact as an octomer with 2:2:2:2 stoichiometry, as supported by gel filtration analysis. The authors use previously developed qPCR assays for several piRNAs to demonstrate that disruption of PICS interactions not only lead to mislocalization of PICS proteins but also to moderate or severe piRNA biogenesis defects. The authors study the essential function of PICS proteins in chromosome segregation, demonstrating separation-of-function effects of some of their single amino acid mutations. Implicated in chromosome segregation are 3 PICS proteins PICS-1/ERH-2/TOST-1 whose mutations lead to sterility (do the authors mean maternal effect embryonic lethality?).

It has previously been suggested that the PICS complex is ~400kd based on size exclusion chromatography (Cordero 2019), suggesting multiple copies of each subunit. Further, human and *pombe* ERH proteins are dimeric, and ERH-2 can self-associate in yeast. The current manuscript provides additional support for the previously proposed model that each subunit of the PICS complex may be present as a dimer. But I am afraid that the current data does not appear to add substantially

to our understanding of piRNA biogenesis or of how RNA metabolism controls cell division.

Comments:

1. A general comment is that this manuscript does not adequately describe previous work on the PICS protein complex. For example, the N-terminal domains of PID-1 and TOST-1 were previously shown to be structurally and functionally homologous based on weak amino acid similarity, identification of a pid-1 mutation in a conserved amino acid in this domain that affects piRNAs, and demonstration that this mutation disrupts 2-hybrid interactions of PID-1 and also that the analogous mutation in TOST-1 disrupts physical interaction with ERH-2. The advances reported in this manuscript will be better understood if the considerable previously published physical and functional interactions are well described.

2. The authors use their own names for some PICS subunits, which in some cases are different from previously published names from another group. For example, PICS-1 is also known as PID-3. The authors state this briefly in the introduction, but then generally use their own gene names. It would be helpful if both published names were used every time a protein is mentioned in the text, for example PICS-1/PID-3. This way it will be easier for readers to assess the significance of previously published work.

3. In line with the above, the authors have previously published that the RRM domain of TOFU-6 interacts with PICS-1 and ERH-2, and also that the C-terminal domain of TOFU-6 interacts with IFE-3. Therefore, several aspects of protein interactions regarding the PICS complex have been previously defined, and it is important to clearly state what has been previously understood and where the current advances are.

4. The calorimetry interactions between the RRM fragments in Fig. 2e-g appear weak or non-cooperative

5. In the Discussion, the authors suggest that pombe Erh2 may be homologous with respect to cell division, but there is little discussion of what this actual function is. In this context, are the RRM domains meant to recruit mitosis-specific RNAs for translation during cell division. If so, could the authors please contrast or compare their results that two RRM domains interact with each with the concept that these RRM domains may bind RNA, at least prior to or after the RRM-based PICS complex is established.

6. In the discussion, the authors state that 'unexpectedly, PICS-1 and TOFU-6 interact with each other via RRM domains, which were previously identified as RNA binding modules'. The authors are correct that RRM domains are known to bind RNA, but I was puzzled by the conclusion that the interaction between PICS-1 and TOFU-6 via RRM domains is unexpected, when this has been demonstrated experimentally in the Cordiero paper on PETISCO/PICS-1.

7. The authors do show that ERH-2 is likely to act as a dimer when it interacts with TOST-1 or PID-1. This speaks to the potential architecture of the PICS complex, but does not address function in a deep manner.

8. In Fig. 5a, the authors state that TOFU-6 interacts at perinuclear granules consistent with previous work. A better description of their images is that TOFU-6 is enriched in perinuclear granules but is also present in cytoplasm and to a lesser degree in the nucleus. It also appears that TOFU-6 is not evenly distributed in the nucleus, possibly present within substructures within the nucleus. The functional significance of these varied levels of PICS proteins in the cytoplasm and nucleus is unclear.

9. Similarly, the description of the text suggests that 'for cells with tofu-6 mutants, PICS-1 failed to

accumulate at the perinuclear granules in the germline, but changed the localization from the cytoplasm to the nucleus'. If one looks at the image this is too a degree true, but some PICS-1 remains in the cytoplasm and the TOFU-6 in the nucleus is apparently excluded from a large part of the nucleus, possibly the nucleolus.

10. Interestingly, TOST-1 is cytoplasmic but becomes primarily nuclear if *erh-2* is mutant, and this is true for TOST-1 mutants with mutations in amino acids that are predicted to disrupt the TOST-1/ERH-2 interaction.

11. The authors are missing a summary figure with a model of what they previously established in their paper on the PICS complex - where 4 proteins were identified and shown to be required for piRNA biogenesis and cell division in Zheng et al, Cell Reports 2019 (model A), and also what was established by the Ketting group in Cordiero 2019 (model B). Compare these published results with what has been learned in this paper (model C). Otherwise, this makes it difficult for the reader to determine how much novel information is presented here. For example, the authors mention IFE-3 in the Discussion, but it is not clear how this translation initiation factor was implicated in piRNAs and what it's relationships to the PICS complex is. The authors could point to a role for IFE-3 in binding the 5' cap of piRNA, which might allow them to speculate on other structural relationships.

12. It would be helpful to clearly discuss previous models about TOST-1 and PID-1: that TOST-1 may antagonize piRNA biogenesis by participating in a parallel pathway that promotes an essential function related to cell division or spliced leader maturation.

13. Previous mass spec and 2 hybrid data revealed that IFE-3 and TOFU-6 interact, that TOFU-6 and PID-3 interact, and that PID-3 interacts with ERH-2. Further, PID-1 and TOST-1 interact with ERH-2. Further, domains of proteins in the PICS PETSICO complex showed that the C-terminal IFE-4 interaction motif of TOFU-6 interacts with IFE-3, that the RRM domain of TOFU-6 interacts with the RRM of PID-3, that the RRM domain of PID-3 interacts with ERH-2. Given the relevance, a summary of such observations would be helpful.

Minor:

1. Abstract 'C. elegans PICS complex plays important roles in piRNA biogenesis, cell division and chromosome segregation, respectively'.

What does respectively refer to here?

2. 'The piRNAs have a precise length of 21 nt, starting with a 5' monophosphorylated uracil'

Usually starting with a 5' uracil

3. 'ITC binding data indicate that the RRM domain'

Please define RRM when this is first used.

4. One subunit of the PICS complex, ERH-2, has homologs in other organisms including fungi. The authors conclude that two PICS proteins PICS-1 and ERH-2 serve as scaffolds for this PICS complex. One question is if *S. pombe* ERH-2 mutants are known to be wildtype of siRNAs or if a population of siRNAs might be altered, similar to the piRNAs of metazoans.

Revision summary

We thank the reviewers for their amazing comments and suggestions. We have conducted a number of new experiments and data analysis to address their concerns and revised the manuscript as the reviewers suggested:

1. added the images of ERH-2::GFP in *pics-1* mutant in Figure 5d.
2. added the images of ERH-2(M38D) in Figure 5e, the piRNA probe levels of ERH-2(M38D) in Figure 7a, the H2B assay of ERH-2(M38D) in Figure 6c, and brood size of ERH-2(M38D) in Supplementary Figure 13.
3. added the images of PID-1(R61C)::GFP in Supplementary Figure 12.
4. We performed additional gel filtration experiments for PICS wild type and mutants, to indicate that mutations within ERM and RRM of PICS-1 impaired the binding to ERH-2 and TOFU-6, respectively, which was also included in Supplementary Fig. 8 of the revised manuscript.
5. We added one model as Fig. 7b to summarize our findings.

For the convenience of examination, we have highlighted all the changes in red in the revised text.

Reviewer #1 (Remarks to the Author):

Wang et al., revealed the molecular basis for PICS complex assembly by using combined structural and biochemical methods. They also identified disruption of these interactions in vivo will lead to mislocalization of PICS subunits, piRNAs decreasing, or abnormal chromosome segregation in C. elegans. Overall, the data are of solid quality. However, there are some concerns which should be addressed before considering publication.

Response: We thank the reviewer for his/her positive comments on our study. We have translated his/her great input and specific points into a well-versed revision.

1. One of the main defects is that there are many misspellings in the text and in the Figures, making it difficult to read the article smoothly. The labelling in the Figures should be improved and be consistent with the text. For examples, Fig 3b (change PICS-1 to EBM as stated in the text) and so on. I also suggest to label Fig 1a with numbers to indicate the domain boundaries clearly as seen in the ITC tables.

Response: We further polished the manuscript by removing the misspellings. We changed the labelling in Fig. 3b from PICS-1 to EBM, and show the numbers in Fig. 1a to indicate the domain boundaries.

2. Another shortcoming is the level of signal to noise is really low for the fluorescent images, making it hard to distinguish proteins localizations.

Response: TOFU-6, PICS-1 and ERH-2 are localized both in cytosol and in the granules. The signals in the cytosol are actually not noise. We have revised the main text to include this information.

3. The article has no information about the PID-1 structure, it just showed PID-1 and TOST-1 mutually exclude each other from the PICS complex by ITC. My understanding is this article mainly focus on the TOST-1 containing complex. So the authors should focus on the chromosome segregation function instead of piRNA biogenesis. Regarding the impact of interaction network on subcellular localizations, I suggest the authors separate the piRNA biogenesis function with the chromosome segregation, instead of mixing them together.

Response: Thanks for the suggestion. In the revised manuscript, we moved old Fig. 5g out to generate new Fig. 7a. In this way, the manuscript focuses more on the chromosome segregation function, and the piRNA biogenesis function is separated from that of chromosome segregation.

4. Does the author have evidence to indicate that ERH-2 binding with PICS-1 will not affect the binding between PICS-1 with TOFU-6 using only RRM+EBM domain?

Response: Thanks for pointing it out. We ran gel filtration experiments by using PICS1⁵⁻²⁸² and its variant. PICS1⁵⁻²⁸², containing N-terminal MID domain, well represents almost full-length PICS-1. In the revised manuscript, we made a new Supplementary Fig. 8 to replace the old figure to indicate the interactions between

PICS subunits. As shown in Supplementary Fig. 8, PICS-1⁵⁻²⁸² interacts with TOFU-6^{RRM} and ERH-2 in a 2:2:2 ratio. In contrast, PICS-1⁵⁻²⁰⁰ (PICS-1^{ΔRRM}), which lacks the RRM domain, binds to ERH-2 but fails to bind TOFU-6^{RRM}. PICS^{ΔEBM}, representing PICS-1⁵⁻²⁸² with EBM (180-200) deleted, binds to TOFU-6^{RRM}, but not ERH-2. In this way, our data confirmed that PICS-1 utilizes EBM and RRM to bind to ERH-2 and TOFU-6, respectively.

Our data is also consistent with previous work by Ketting's work and ours, which independently demonstrated by pull-down and genetic experiments that full length PICS-1/PID-3 binds to ERH-2 and TOFU-6 via the EBM-RRM region (PMID: 31147388, Fig. 3; PMID: 31216475, Fig. S11D).

5. *The authors should discriminate PIWI (subfamily name) and Piwi (protein name) throughout the text.*

Response: Thanks the reviewer's suggestion. Accordingly, we used PIWI for subfamily name and Piwi for protein name in the text.

6. *Author using a buffer containing 200mM NaCl for ITC analysis, is there any specific reason to use this high salt buffer?*

Response: Thanks for pointing it out. We tried different salt concentration, 200 mM NaCl is the optimal one to keep protein stable during ITC titration.

Reviewer #2 (Remarks to the Author):

This manuscript provides the structural basis for molecular interactions in PICS complex composed of TOFU-6, PICS-1, ERH, and either of PID-1 or TOST-1, which is crucial for piRNA biogenesis and chromosome segregation/cell division, respectively. The authors solved the high-resolution crystal structures of the complexes of PICS-1(RRM) and TOFU-6 (RRM), of ERH-2 and PICS-1 fragment, and of TOST-1 fragment and ERF-2, and the atomic interactions are confirmed by ITC experiments of mutants. Moreover, they analyzed piRNA biogenesis and localization/cell division of the structure-designed mutants. While the structure determinations are solid, and the biochemical and cell biological results have impact to some extent, this reviewer has some concerns before acceptance of this paper.

Response: We thank this reviewer for his/her positive comments on our manuscript.

1. *As this manuscript does not provide any results of RNA-binding and DNA(chromatin) binding results in the structural viewpoint, this reviewer feels fluctuation that the structural results and the functional analyses seem separated. Therefore, this reviewer would like to ask the authors to provide some structural data of interaction between PICS complex and nucleic acids.*

Response: Thanks for the reviewer's suggestion. By solving several PICS subcomplex structures, our work provides conceptual advance in several aspects: 1) PICS-1^{RRM} forms a homodimer and binds to TOFU-6^{RRM} in a ratio of 2:2. 2) ERH-2 homodimer binds to PICS-1^{EBM} in a ratio of 2:2. Intriguingly, the ERH-2 homodimer

binds to PICS-1^{EBM-RRM} homodimer in a ratio of 1:1, rather than forming aggregated polymers (Fig. 1 for reviewers). 3) ERH-2 homodimer utilizes different surface to bind to PICS-1^{EBM} and TOST-1/PID-1. The mutations impairing the intact PICS complex lead to mis-localization of PICS subunits and defects in cell division and/or piRNA biogenesis. We also summarized our work as a proposed model in Fig. 7b of the revised manuscript.

Fig. 1 for reviewers. ERH-2 homodimer binds to PICS-1 homodimer in a ratio of 1:1.

We also compared the PICS-1 homodimer with the other two solved RRM homodimers, MEC-8 and HuR, and found that both RRM homodimers bind to the RNA via the canonical site. Accordingly, we added a new figure (Supplementary Fig. 4) in the result section to compare RRM domain homodimers and to indicate potential RNA binding sites for RRM domains of PICS-1 and TOFU-6. We also added one paragraph in the revised manuscript as follows:

“Homodimerization of PICS-1RRM is similar to those observed for the MEC-8 (PDB ID: 5TKZ) and HuR (PDB ID: 6GC5) RRM homodimers, both of which homodimerize via the $\alpha 1$ helix and have the RNA binding capacity (Supplementary Fig. 4). The RNA-bound complex structures demonstrate that MEC-8 RRM and HuR RRM bind their nucleotide ligands in a way common for other known RRM domains (Supplementary Fig. 4c-4f). Of note, the corresponding surfaces of PICS-1 RRM and TOFU-6 RRM are also positively charged (Supplementary Fig. 4a-4b), suggesting both of them likely employ similar RNA binding mode.”

2. *The authors presented the separate complex structures, but they describe that they could purify the complex consisting of ERH-2-(GSS)6-TOST-1, PICS-1(EBM-RRM), and TOFU-6(RRM) in gel filtration. Why did not they solve the complex structure by X-ray crystallography or Cryo-EM single particle analysis ?*

Response: Thanks for the reviewer’s suggestion. We tried the crystallization of the complex but failed to get any crystals. Also the molecular weight of the complex is relative small for cryo-EM. Although we did not solve the structure of PICS complex, the presented subcomplex structures support our conclusion by uncovering the molecular mechanism underlying intermolecular interactions within PICS.

3. *The authors should pay more attention to prepare the manuscript. The followings are their mistakes:*

- 1) In the first line of the introduction section, paly should be play.
- 2) In the second line of page 6, "homodimer" is duplicated.
- 3) In line 11 on the same page, they should add "and the side chain of Arg227B" just after Met213A.

Response: They were all corrected in the revised manuscript.

Reviewer #3 (Remarks to the Author):

The authors and an independent group have previously identified the PICS complex, also known as PETISCO, and both previous papers arrived at remarkably consistent conclusions: that multiple components of PICS play roles in piRNA biogenesis, which is not essential for development or fertility, and in a distinct process that is essential for embryo viability. The previous studies identified two 'adaptor' proteins that mediate these functions - PID-1, which promotes piRNA biogenesis in conjunction with the 5' cap-binding factor IFE-3, and TOST-1, which inhibits piRNA function and is also essential for embryo development (chromosome segregation). It has previously been suggested that PID-1 and TOST-1 function in substrate recognition or processing. PID-1 and TOST-1 were previously shown to share a motif that binds to ERH-2 in a mutually exclusive manner. It was previously shown that the RRM domains of TOFU-6 and PICS-1 interact. The current manuscript presents detailed structural and functional analysis of several domains in PICS complex protein, in vitro and in vivo.

In the current study, the authors use elegant structural biology and biochemistry analysis to determine that the RRM domains of TOFU-6 and PICS-1 form homodimers that interact as a heterotetramer. Second, the authors show that ERH-2 has two distinct domains that interact with PID-1 or TOST-1, although these interactions are competitive and mutually exclusive. As with the RRM domains, the interactions between ERH-2 and PID-1 or TOST-1 are mutually exclusive. The major insight provided by the current manuscript is evidence for the stoichiometry of the PICS complex, where each subunit is present in two copies per PICS complex. The localization and function of PICS subunits where critical interacting amino acids are mutated is determined. This provides some insight into the subcellular distribution of PICS subunits. That said, there do not appear to be major new insights regarding either the piRNA maturation or the cell division functions of PICS subunits, apart from linking perinuclear granule localization to piRNA biogenesis, a concept that has already been well established in the field.

For example, in the Discussion, the authors point to previous ERH structures for pombe and human that are dimers, which is consistent with their model that PICS subunits are present in two copies per PICS complex. Beyond the presence of dimers of each subunit, it is not clear what this teaches us about PICS function. The combined structural biology and cell biology do not appear to provide significant insight into PICS function.

Response: We thank this reviewer for his/her thoughtful suggestions and the positive comments on our manuscript. By solving several PICS subcomplex structures, our work provides conceptual advance in several aspects: 1) PICS-1 homodimerizes via the RRM domain, which is a novel finding and its molecular mechanism was uncovered by structural biology. 2) ERH-2 is a homodimer conserved from yeast to human. In this way, two binding modes are possible for the PICS-1 homodimer binding to the ERH-2 homodimer. One is that different homodimers form heterogeneous aggregation, the other is that the two homodimers are assembled accurately in a 1:1 ratio. Our gel filtration experiments unambiguously supported the latter. Therefore, the precise assembly between ERH-2^{EBM-RRM} and ERH-2 constitutes the basis for the homogeneous PICS octamer (**Fig. 1 for reviewers**). 3) ERH-2 homodimer also utilizes different surface to bind to PICS-1^{EBM} and TOST-1/PID-1. The mutations impairing the intact PICS complex lead to mislocalization of PICS subunits and defects in cell division and/or piRNA biogenesis.

Fig. 1 for reviewers. ERH-2 homodimer binds to PICS-1 homodimer in a ratio of 1:1.

Overall, we made a new figure in the revised manuscript as Fig. 7b, to summarize the conceptual advance provided by structural biology.

Fig. 7b. Proposed model for dual roles of PICS complex.

The authors solve crystal structures of domains of PICS proteins, which leads them to predict that the four proteins interact as an octomer with 2:2:2:2 stoichiometry, as supported by gel filtration analysis. The authors use previously developed qPCR assays for several piRNAs to demonstrate that disruption of PICS interactions not only lead to mislocalization of PICS proteins but also to moderate or severe piRNA biogenesis defects. The authors study the essential function of PICS proteins in chromosome segregation, demonstrating separation-of-function effects of some of their single amino acid mutations. Implicated in chromosome segregation are 3 PICS proteins PICS-1/ERH-2/TOST-1 whose mutations lead to sterility (do the authors mean maternal effect embryonic lethality?).

Response: Thanks for the comments. It is maternal effect embryonic lethality. We have revised the main text to include this information.

It has previously been suggested that the PICS complex is ~400kd based on size exclusion chromatography (Cordiero 2019), suggesting multiple copies of each subunit. Further, human and pombe ERH proteins are dimeric, and ERH-2 can self-associate in yeast. The current manuscript provides additional support for the previously proposed model that each subunit of the PICS complex may be present as a dimer. But I am afraid that the current data does not appear to add substantially to our understanding of piRNA biogenesis or of how RNA metabolism controls cell division.

Response: We thanks the reviewer for pointing it out. Cordiero *et al.* did identify the ~400 kD peaks for PICS complex, our work further uncovered that the basis of dimerization come from both PICS-1^{RRM} and ERH-2 and the interaction network underlying it. To acknowledge it, we added one sentence with the citation in the introduction section as follows:

“In addition, PICS was purified from cell extracts as a complex of ~400 kD, suggesting the possibility of multiple copies of each subunit³⁸.”

Comments:

1. A general comment is that this manuscript does not adequately describe previous work on the PICS protein complex. For example, the N-terminal domains of PID-1 and TOST-1 were previously shown to be structurally and functionally homologous based on weak amino acid similarity, identification of a pid-1 mutation in a conserved amino acid in this domain that affects piRNAs, and demonstration that this mutation disrupts 2-hybrid interactions of PID-1 and also that the analogous mutation in TOST-1 disrupts physical interaction with ERH-2. The advances reported in this manuscript will be better understood if the considerable previously published physical and functional interactions are well described.

Response: Thanks for the suggestion. We have revised the discussion section to discuss the previously published physical and functional interactions. It is shown as follows:

“Consistent with previous work^{38, 39}, PICS-1(PID-3) and TOFU-6 interact with

each other via RRM domains, which were previously identified as the RNA binding modules. PICS-1 binds to TOFU-6 and ERH-2 via RRM and EBM, respectively. ERH-2 and PICS-1(PID-3) are both homodimers within the PICS complex and the two homodimers interact with each other in a 1:1 ratio to form a homogenous tetramer rather than high-order polymers or aggregates, which constitutes the basis of PICS octamer (**Fig. 7b**).”

In addition, we constructed GFP tagged PID-1(R61C) transgene, a mutation that disrupts the interaction between ERH-2 and PID-1 and reduced piRNA production (PMID: 24696453; PMID: 31147388). The expression of PID-1(R61C)::GFP is very low, consistent with previous Western blot assay (PMID: 24696453, Figure 3A) and similar to those observed for TOST-1(R42C)::GFP in germ-cells (Supplementary Fig. S12).

2. The authors use their own names for some PICS subunits, which in some cases are different from previously published names from another group. For example, PICS-1 is also known as PID-3. The authors state this briefly in the introduction, but then generally use their own gene names. It would be helpful if both published names were used every time a protein is mentioned in the text, for example PICS-1/PID-3. This way it will be easier for readers to assess the significance of previously published work.

Response: We used PICS-1/PID-3 throughout the revised manuscript.

3. In line with the above, the authors have previously published that the RRM domain of TOFU-6 interacts with PICS-1 and ERH-2, and also that the C-terminal domain of TOFU-6 interacts with IFE-3. Therefore, several aspects of protein interactions regarding the PICS complex have been previously defined, and it is important to clearly state what has been previously understood and where the current advances are.

Response: We have revised the main text as follows to include this information: “Although IFE-3, which encodes the ortholog of human eIF4E, binds to the C-terminal region of TOFU-6, named as IFE-3 binding motif (IBM) hereafter^{38,39}, its depletion does not affect the localization of the other PICS subunits³⁹. Previous work indicated that the PICS-1 (also known as PID-3) binds to the RNA-Recognition Motif (RRM) of TOFU-6 and ERH-2, and ERH-2 further binds to TOST-1 or PID-1^{38,39}. In addition, PICS was purified from cell extracts as a complex of ~400kD, suggesting the possibility of multiple copies of each subunit³⁸. However, the molecular mechanisms underlying the interaction network within PICS complex are largely unknown.”

4. The calorimetry interactions between the RRM fragments in Fig. 2e-g appear weak or non-cooperative

Response: To be consistent with the binding curve shown in Fig. 1b, we used the same scale in Figs. 2e-2g. Actually the heat released from the interactions for mutants

are still good enough to obtain K_d values. Here we rescaled the y axis to show the curve more clearly as below:

Fig. 2 for reviewers. y axis rescaled for Figs. 2e-2g to show ITC binding curves.

5. In the Discussion, the authors suggest that *pombe* Erh2 may be homologous with respect to cell division, but there is little discussion of what this actual function is. In this context, are the RRM domains meant to recruit mitosis-specific RNAs for translation during cell division. If so, could the authors please contrast or compare their results that two RRM domains interact with each with the concept that these RRM domains may bind RNA, at least prior to or after the RRM-based PICS complex is established.

Response: Thanks the reviewer for the suggestion.

Firstly, we compared the PICS-1 homodimer with the other two solved RRM homodimers, MEC-8 and HuR, and found that both RRM homodimers bind to the RNA via the canonical site. Accordingly, we added a new figure (Supplementary Fig. 4) in the result section to compare RRM domain homodimers and to indicate potential RNA binding sites for RRM domains of PICS-1 and TOFU-6. We also added one paragraph in the revised manuscript as follows:

“Homodimerization of PICS-1RRM is similar to those observed for the MEC-8 (PDB ID: 5TKZ) and HuR (PDB ID: 6GC5) RRM homodimers, both of which homodimerize via the $\alpha 1$ helix and have the RNA binding capacity (Supplementary Fig. 4). The RNA-bound complex structures demonstrate that MEC-8 RRM and HuR RRM bind their nucleotide ligands in a way common for other known RRM domains (Supplementary Fig. 4c-4f). Of note, the corresponding surfaces of PICS-1 RRM and TOFU-6 RRM are also positively charged (Supplementary Fig. 4a-4b), suggesting both of them likely employ similar RNA binding mode.”

Next we revised the discussion section to discuss the possible function of ERH-2 orthologs, including *pombe* Erh2 and human ERH, with appropriate citations, which are as follows:

“Although the binding ligands for ERH-2 and its orthologs are not conserved, *Schizosaccharomyces pombe* Erh1 and human ERH were also reported to be

associated with cell division⁴³. Specifically, Erh1-Mmi1 plays an important role in degrading specific transcripts during meiosis⁴⁴, whereas mammalian ERH is indispensable for chromosome segregation during mitosis because of its key role in regulating the expression of CENP-E, a centromere-associated kinesin-like motor protein^{45, 46}. The diverse functions of mammalian ERH are attributed to its capacity of binding with different ligands, such as DGCR8, POLDIP3, and CIZ1, *etc*⁴⁴. Similarly, the role of *C. elegans* ERH-2 in mitosis could also be relevant to its potential nuclear function in gene regulation via binding to TOST-1 and other unknown factors. Furthermore, by binding to TOST-1 and PID-1 in a mutual exclusive manner, ERH-2 might act as an anchor protein not only in conferring TOST-1 the ability to antagonize piRNA biogenesis, but also establishing a possible spatial and temporal crosstalk between two biological events.”

6. *In the discussion, the authors state that ‘unexpectedly, PICS-1 and TOFU-6 interact with each other via RRM domains, which were previously identified as RNA binding modules’. The authors are correct that RRM domains are known to bind RNA, but I was puzzled by the conclusion that the interaction between PICS-1 and TOFU-6 via RRM domains is unexpected, when this has been demonstrated experimentally in the Cordiero paper on PETISCO/PICS-1.*

Response: Thanks the reviewer for pointing it out. We changed the statement as follows with added citations: “Consistent with previous reports, PICS-1 and TOFU-6 interact with each other via RRM domains, which were previously identified as the RNA binding modules.”

7. *The authors do show that ERH-2 is likely to act as a dimer when it interacts with TOST-1 or PID-1. This speaks to the potential architecture of the PICS complex, but does not address function in a deep manner.*

Response: Thanks the reviewer for the comment. The ERH-2 homodimer is conserved from human to yeast, and it also constitutes the basis of PICS octamer. However, the diverse biological function of ERH-2 homodimer and its orthologs as a scaffold protein, is really enigmatic. By comparing the similar role of its human ortholog in mitosis, we also proposed that the function of ERH-2 in cell division may be attributed to its potential role in gene regulation, although further work might be required to uncover its unknown factors.

We revised the discussion section to discuss the possible function of ERH-2 orthologs, including pombe Erh2 and human ERH, with appropriate citations. It is as follow:

“Although the binding ligands for ERH-2 and its orthologs are not conserved, *Schizosaccharomyces pombe* Erh1 and human ERH were also reported to be associated with cell division⁴³. Specifically, Erh1-Mmi1 plays an important role in degrading specific transcripts during meiosis⁴⁴, whereas mammalian ERH is indispensable for chromosome segregation during mitosis because of its key role in regulating the expression of CENP-E, a centromere-associated kinesin-like motor protein^{45, 46}. The diverse functions of mammalian ERH are attributed to its capacity of

binding with different ligands, such as DGCR8, POLDIP3, and CIZ1, *etc*⁴⁴. Similarly, the role of *C. elegans* ERH-2 in mitosis could also be relevant to its potential nuclear function in gene regulation via binding to TOST-1 and other unknown factors. Furthermore, by binding to TOST-1 and PID-1 in a mutual exclusive manner, ERH-2 might act as an anchor protein not only in conferring TOST-1 the ability to antagonize piRNA biogenesis, but also establishing a possible spatial and temporal crosstalk between two biological events.”

8. *In Fig. 5a, the authors state that TOFU-6 interacts at perinuclear granules consistent with previous work. A better description of their images is that TOFU-6 is enriched in perinuclear granules but is also present in cytoplasm and to a lesser degree in the nucleus. It also appears that TOFU-6 is not evenly distributed in the nucleus, possibly present within substructures within the nucleus. The functional significance of these varied levels of PICS proteins in the cytoplasm and nucleus is unclear.*

Response: We revised the main text as suggested to better describe the subcellular location of TOFU-6. TOFU-6 does not go to the nucleus in wild type and various mutant background.

9. *Similarly, the description of the text suggests that ‘for cells with tofu-6 mutants, PICS-1 failed to accumulate at the perinuclear granules in the germline, but changed the localization from the cytoplasm to the nucleus’. If one looks at the image this is too a degree true, but some PICS-1 remains in the cytoplasm and the TOFU-6 in the nucleus is apparently excluded from a large part of the nucleus, possibly the nucleolus.*

Response: We revised the main text as suggested to better describe the subcellular location of PICS-1/PID-3. TOFU-6 does not go to the nucleus in wild type and various mutant background.

10. *Interestingly, TOST-1 is cytoplasmic but becomes primarily nuclear if erh-2 is mutant, and this is true for TOST-1 mutants with mutations in amino acids that are predicted to disrupt the TOST-1/ERH-2 interaction.*

Response: Thanks for pointing it out. We agreed with the reviewer that it is quite interestingly why mutation of some factors or disrupting their interaction will change the stability or subcellular localization of the others. We are still at the very early stage to investigate the mechanism. We have revised the discussion to include this information.

11. *The authors are missing a summary figure with a model of what they previously established in their paper on the PICS complex - where 4 proteins were identified and shown to be required for piRNA biogenesis and cell division in Zheng et al, Cell Reports 2019 (model A), and also what was established by the Ketting group in Cordiero 2019 (model B). Compare these published results with what has been learned in this paper (model C). Otherwise, this makes it difficult for the reader to*

determine how much novel information is presented here. For example, the authors mention IFE-3 in the Discussion, but it is not clear how this translation initiation factor was implicated in piRNAs and what its relationships to the PICS complex is. The authors could point to a role for IFE-3 in binding the the 5' cap of piRNA, which might allow them to speculate on other structural relationships.

Response: Thanks for the suggestion.

Firstly, we added one sentence in the introduction section to indicate IFE-3 is associated with PICS via TOFU-6 as follows:

“Although IFE-3, which encodes the ortholog of human eIF4E, binds to the C-terminal region of TOFU-6^{38,39}, its depletion does not affect the localization of the other PICS subunits³⁹.”

Secondly, we modified Fig. 1a by indicating the IFE-3 binding motif (IBM) at the C-terminus of TOFU-6.

Thirdly, in the discussion section, we further discussed the relationship of IFE-3 and PICS subunits by pointing out that IFE-3 is a 5' cap binding protein and its potential role in recruiting PICS. We added one paragraph as follows:

“Previously, we and Ketting group have independently identified IFE-3 associated with TOFU-6, PICS-1(PID-3), ERH-2, PID-1 and TOST-1 from IP-MS, and found that recombinant IFE-3 directly binds to the IBM motif of TOFU-6^{38,39}. IFE-3 acts as the binding protein for 5' capped piRNAs and is enriched in P-granule^{38, 39}. Also *ife-3* mutant leads to defects in piRNA biogenesis and maternal effect lethality³⁸. However, more evidence imply that the functions of IFE-3 do not fully overlap with those of PICS complex. Firstly, IFE-3 but not PICS, binds directly to capped piRNA precursors³⁸. Secondly, *tofu-6* mutant worms, but not *ife-3* null mutant worms, have germline and eggs, suggesting that *ife-3* null mutant is more severe. Thirdly, IFE-3 accumulates in peri-nuclear granules in germ-cells and embryos, whereas TOFU-6 is localized in granules in germ-cells but not in embryos (**Fig. 3 for reviewers**)^{38,39}. Fourthly, unlike the mutual dependence of PICS factors, the perinuclear localization of TOFU-6 is independent of *ife-3* (Zeng, 2019 Cell Reports). Fifthly, although the role of IFE-3 in piRNA biogenesis was reported, whether it functions in cell division is largely unknown.”

Fig. 3 for reviewers. Localizations of IFE-3 and TOFU-6 in germline and embryos.

12. It would be helpful to clearly discuss previous models about TOST-1 and PID-1: that TOST-1 may antagonize piRNA biogenesis by participating in a parallel pathway that promotes an essential function related to cell division or spliced leader maturation.

Response: Thanks for the suggestion. We have revised the discussion section to clearly describe the previous model about TOST-1 and PID-1. We added one paragraph in the discussion section as follow:

“In contrast, the interactions between PICS-1(PID-3), ERH-2 and TOST-1 are essential in chromosome segregation during cell division. Therefore, PICS mediates cell division via the PICS-1/ERH-2/TOST-1 axis, in which ERH-2 dimer acts as the scaffold to tether PICS-1(PID-3) and TOST-1. In addition, TOST-1 is cytoplasmic but becomes primarily nuclear in *erh-2* mutant, and this is also true for TOST-1 mutants that disrupt the TOST-1/ERH-2 interaction. It is likely that ERH-2/TOST-1 interaction might be required for maintaining the ERH-2 level in the nucleus, which might be essential for cell division or spliced leader maturation.”

13. Previous mass spec and 2 hybrid data revealed that IFE-3 and TOFU-6 interact, that TOFU-6 and PID-3 interact, and that PID-3 interacts with ERH-2. Further, PID-1 and TOST-1 interact with ERH-2. Further, domains of proteins in the PICS PETSICO complex showed that the C-terminal IFE-4 interaction motif of TOFU-6 interacts with IFE-3, that the RRM domain of TOFU-6 interacts with the RRM of PID-3, that the RRM domain of PID-3 interacts with ERH-2. Given the relevance, a summary of such observations would be helpful.

Response: Thanks for the suggestion. We have included a model figure in the revised manuscript (Fig. 7b) and summarized current work in the discussion section. We have also revised the discussion section about the function of IFE-3 in piRNA biogenesis by adding a paragraph to discuss in detail the relationship between IFE-3 and PICS, as shown in the response to comment No. 11. In the introduction section, we also summarized the findings from previous work by Ketting group and us, which is shown as follow:

“Previous work indicated that the PICS-1 (also known as PID-3) binds to the RNA-Recognition Motif (RRM) of TOFU-6 and ERH-2, and ERH-2 further binds to TOST-1 or PID-1^{38,39}. In addition, PICS was purified from cell extracts as a complex of ~400 kD, suggesting the possibility of multiple copies of each subunit³⁸.”

Minor:

1. Abstract “*C. elegans* PICS complex plays important roles in piRNA biogenesis, cell division and chromosome segregation, respectively’.

What does respectively refer to here?

Response: It was corrected as “*C. elegans* PICS complex plays important roles in piRNA biogenesis, cell division, and chromosome segregation.”

2. *'The piRNAs have a precise length of 21 nt, starting with a 5' monophosphorylated uracil'*

Usually starting with a 5' uracil

Response: It was corrected as "5' uracil".

3. *'ITC binding data indicate that the RRM domain'*

Please define RRM when this is first used.

Response: It was defined in the introduction section as "ITC binding data indicate that the RNA-Recognition Motif (RRM) domain and a fragment upstream of the RRM domain of PICS-1/PID-3 interact with the TOFU-6 RRM domain and ERH-2, ..."

4. *One subunit of the PICS complex, ERH-2, has homologs in other organisms including fungi. The authors conclude that two PICS proteins PICS-1 and ERH-2 serve as scaffolds for this PICS complex. One question is if S. pombe ERH-2 mutants are known to be wildtype of siRNAs or if a population of siRNAs might be altered, similar to the piRNAs of metazoans.*

Response: In *S. pombe*, Erh1, in association with Mmi1, was known to play important roles in degrading meiosis-specific transcripts, silencing gametogenic genes, and in regulating long non-coding RNAs (PMID: 31974447; PMID: 30651569; PMID: 32546512; PMID: 26631744). However, Whether Erh1 functions in siRNA pathway remains elusive.

REVIEWERS' COMMENTS

Reviewer #1 (Remarks to the Author):

I'm satisfied with the revised version and my concerns are all answered by the authors.

Reviewer #2 (Remarks to the Author):

In the light of the reply to my concerns, I understand the situation and am satisfied with the revision. Therefore, I am now positive to consider publication of this manuscript in Nature Communications.

Reviewer #3 (Remarks to the Author):

Wang and colleagues present a revision for their manuscript on the structural basis for PICS/ERH2 function. They report crystal structures of PICS-1/TOFU-6 RRM domains, PICS-1 EMB domain with ERH-2, and ERH-2 with a 20 amino acid segment of the cell division protein TOST-1. This provides a structural model from which to consider the chromosome segregation function of the PICS-1/TOFU-6 complex. The authors present in vivo localization that tests several of structural features that they define, which resulted in altered cytoplasmic, nuclear or perinuclear localization of TOFU-6, ERH-2 or TOST-1. In some cases, altered piRNA levels are also observed, as expected for the dual piRNA and cell division functions of TOFU-6 and ERH-2.

That said, the structural biology and cell biology that are presented in this manuscript do not provide substantial new insights into how chromosome segregation or piRNA biogenesis are regulated or orchestrated by PICS-1/TOFU-6 and associated proteins. By contrast, previous publications by Zeng in Cell Reports and by Cordiero in Genes and Development performed a great deal of biochemistry and cell biology to define roles of PICS-1/TOFU-6 and associated proteins in piRNA biogenesis and chromosome segregation. Although subunits of the PICS-1/TOFU-6 complexes such as ERH-2 are conserved in mammals and yeast, the authors do not present a common function that might explain the relationship between ERH-2 why this complex regulates the apparently distinct processes of piRNA biogenesis and chromosome segregation. One possibility is that the chromosome segregation function of TOST-1 is related to biogenesis of CSR-1 small RNAs that are known to be required for chromosome segregation, but this manuscript fails to present a clear model for how PICS-1/TOFU-6 complex proteins promote their biological functions, apart from defining the stoichiometry of the complexes. The authors do create a reasonable framework for understanding some of the structural interactions within PICS-1/TOFU-6 complexes and I wish them well on their future endeavors.

Comment

1. 'While the TOST-1 containing form is essential for chromosome segregation and cell division in embryos, suggesting that a potential crosstalk exists between the two biological events'. It is not clear why crosstalk between chromosome segregation and cell division is being suggested here, as cell division and chromosome segregation are linked events. A broader question is why piRNA biogenesis and cell division should be linked by a common scaffold of proteins, but this manuscript does not address this relationship.

Reviewer #1 (Remarks to the Author):

I'm satisfied with the revised version and my concerns are all answered by the authors.

Reviewer #2 (Remarks to the Author):

In the light of the reply to my concerns, I understand the situation and am satisfied with the revision. Therefore, I am now positive to consider publication of this manuscript in Nature Communications.

Reviewer #3 (Remarks to the Author):

Wang and colleagues present a revision for their manuscript on the structural basis for PICS/ERH2 function. They report crystal structures of PICS-1/TOFU-6 RRM domains, PICS-1 EMB domain with ERH-2, and ERH-2 with a 20 amino acid segment of the cell division protein TOST-1. This provides a structural model from which to consider the chromosome segregation function of the PICS-1/TOFU-6 complex. The authors present in vivo localization that tests several of structural features that they define, which resulted in altered cytoplasmic, nuclear or perinuclear localization of TOFU-6, ERH-2 or TOST-1. In some cases, altered piRNA levels are also observed, as expected for the dual piRNA and cell division functions of TOFU-6 and ERH-2.

That said, the structural biology and cell biology that are presented in this manuscript do not provide substantial new insights into how chromosome segregation or piRNA biogenesis are regulated or orchestrated by PICS-1/TOFU-6 and associated proteins. By contrast, previous publications by Zeng in Cell Reports and by Cordiero in Genes and Development performed a great deal of biochemistry and cell biology to define roles of PICS-1/TOFU-6 and associated proteins in piRNA biogenesis and chromosome segregation. Although subunits of the PICS-1/TOFU-6 complexes such as ERH-2 are conserved in mammals and yeast, the authors do not present a common function that might explain the relationship between ERH-2 why this complex regulates the apparently distinct processes of piRNA biogenesis and chromosome segregation. One possibility is that the chromosome segregation function of TOST-1 is related to biogenesis of CSR-1 small RNAs that are known to be required for chromosome segregation, but this manuscript fails to present a clear model for how PICS-1/TOFU-6 complex proteins promote their biological functions, apart from defining the stoichiometry of the complexes. The authors do create a reasonable framework for understanding some of the structural interactions within PICS-1/TOFU-6 complexes and I wish them well on their future endeavors.

Response: We thank all three reviewers for their valuable suggestions and comments, which greatly contributed to improving the manuscript.

Comment

1. *'While the TOST-1 containing form is essential for chromosome segregation and cell division in embryos, suggesting that a potential crosstalk exists between the two biological events'. It is not clear why crosstalk between chromosome segregation and cell division is being suggested here, as cell division and chromosome segregation are linked events. A broader question is why piRNA biogenesis and cell division should be linked by a common scaffold of proteins, but this manuscript does not address this relationship.*

Response: We changed the sentence to "... while the TOST-1 containing form is essential for chromosome segregation and cell division in embryos. Therefore our work provides a good example by showing that PICS complex adapts to two distinct cellular functions, piRNA biogenesis and cell division, by incorporating two mutually exclusive factors." We agree with this reviewer on that further work is required to uncover how piRNA biogenesis and cell division are linked by PICS or its associated factors.